# Altered hierarchical auditory predictive processing after lesions to the orbitofrontal cortex

**Olgerta Asko[1]\*, Alejandro Omar Blenkmann[1], Sabine Liliana Leske[2], Maja Dyhre Foldal[1], Anais LLorens[3,4,5], Ingrid Funderud[6,7], Torstein R Meling[8], Robert T Knight[3], Tor Endestad[1,6], Anne-Kristin Solbakk[1,6,9]**

[1]RITMO Centre for Interdisciplinary Studies in Rhythm, Time and Motion, Department of Psychology, University of Oslo, Oslo, Norway; [2]RITMO Centre for Interdisciplinary Studies in Rhythm, Time and Motion, Department of Musicology, University of Oslo, Oslo, Norway; [3]Helen Wills Neuroscience Institute and Department of Psychology, University of California, Berkeley, Berkeley, United States; [4]Université de Franche-Comté, SUPMICROTECH, CNRS, Institut FEMTO-ST, Besançon, France; [5]Université Paris Cité, Institute of Psychiatry and Neuroscience of Paris (IPNP), INSERM U1266, Team TURC, Paris, France; [6]Department of Neuropsychology, Helgeland Hospital, Mosjøen, Norway; [7]Regional Department of Eating Disorders, Oslo University Hospital, Oslo, Norway; [8]Department of Neurosurgery, National Hospital, Copenhagen, Denmark; [9]Department of Neurosurgery, Oslo University Hospital, Oslo, Norway

**\*For correspondence:**
olgerta.asko@psykologi.uio.no

**Competing interest:** The authors declare that no competing interests exist.

**Abstract** Orbitofrontal cortex (OFC) is classically linked to inhibitory control, emotion regulation, and reward processing. Recent perspectives propose that the OFC also generates predictions about perceptual events, actions, and their outcomes. We tested the role of the OFC in detecting violations of prediction at two levels of abstraction (i.e., hierarchical predictive processing) by studying the event-related potentials (ERPs) of patients with focal OFC lesions (n = 12) and healthy controls (n = 14) while they detected deviant sequences of tones in a local–global paradigm. The structural regularities of the tones were controlled at two hierarchical levels by rules defined at a local (i.e., *between tones within sequences*) and at a global (i.e., *between sequences*) level. In OFC patients, ERPs elicited by standard tones were unaffected at both local and global levels compared to controls. However, patients showed an attenuated mismatch negativity (MMN) and P3a to local prediction violation, as well as a diminished MMN followed by a delayed P3a to the combined local and global level prediction violation. The subsequent P3b component to conditions involving violations of prediction at the level of global rules was preserved in the OFC group. Comparable effects were absent in patients with lesions restricted to the lateral PFC, which lends a degree of anatomical specificity to the altered predictive processing resulting from OFC lesion. Overall, the altered magnitudes and time courses of MMN/P3a responses after lesions to the OFC indicate that the neural correlates of detection of auditory regularity violation are impacted at two hierarchical levels of rule abstraction.

## Editor's evaluation

This important study demonstrates that the orbitofrontal cortex is involved in the detection of local auditory prediction errors. The methods and procedures are convincing, although the precise functional meaning of the reported effects remains to be specified. This work will be of interest

to neuropsychologists and cognitive neuroscientists working on the prefrontal cortex, predictive processing, auditory perception, and electrophysiology.

## Introduction

The orbitofrontal cortex (OFC) is linked to multiple high-level cognitive processes, including inhibitory control (*Godefroy et al., 1999*; *Harlow, 1993*; *Picton et al., 2007*), goal-directed attention (*Badre and Wagner, 2004*; *Luu et al., 2000*; *Walton et al., 2004*), temporal context memory (*Duarte et al., 2010*; *Shimamura et al., 1990*), and working memory (WM) maintenance (*Llorens et al., 2019*; *Barbey et al., 2011*; *D'Esposito et al., 2000*), all supporting pivotal involvement of the OFC in decision-making (*Bechara et al., 2000a*; *Ernst et al., 2002*; *Hebscher and Gilboa, 2016*; *Ullsperger and von Cramon, 2004*). Extensive research has highlighted a key role of the OFC in outcome monitoring and evaluation (e.g., *Walton et al., 2004*; *Ullsperger and von Cramon, 2004*; *Wallis, 2012*), and the signaling of reward value (*Howard and Kahnt, 2021*; *Schoenbaum et al., 2009*; *Schoenbaum et al., 2011*; *Wikenheiser and Schoenbaum, 2016*). Patients with OFC damage typically demonstrate difficulty in learning from previous errors, often combined with reduced sensitivity to future consequences (*Bechara et al., 2000a*; *Bechara et al., 2000b*; *Floden et al., 2008*; *Stuss et al., 2000*). However, studies using tasks where outcomes do not have a clear emotional or motivational value support the idea that the OFC forms representations of the environment that extend beyond reward (*Schoenbaum et al., 2011*; *McDannald et al., 2014*; *Stalnaker et al., 2014*). A recent proposal is that OFC generates specific predictions about impending events, such as their identity and features, and uses relevant contextual and temporal attributes to allow continuous updating of rules (*Schuck et al., 2016*; *Stalnaker et al., 2015*; *Wilson et al., 2014*). This emerging view of OFC in monitoring the environment requires predicting events and outcomes but also noticing violations of expectation, that is, prediction errors (PEs) (*Petrides, 2007*). Notably, animal (*Sul et al., 2010*; *Thorpe et al., 1983*; *Wallis and Rich, 2011*) and human (*Nobre et al., 1999*; *Schultz and Dickinson, 2000*; *Tobler et al., 2006*) studies have reported a role of the OFC in signaling PEs, but its involvement in the generation of PEs at different levels of content and temporal complexity remains largely unknown.

Noticing violations of expectation may occur across all levels of extended brain systems in a hierarchical manner, such that higher-level structures predict inputs from lower-level ones through top-down connections, and error signals are sent back via bottom-up connections to update the current model of the environment (*Bar, 2009*; *Clark, 2013*; *Friston, 2010*; *Rao, 2005*). According to the predictive coding hypothesis, the brain continually formulates predictions about sensory inputs and tests them against incoming sensory signals, wherein the bottom-up information represents the interaction between the prediction and the actual sensory input (i.e., PE) (*Clark, 2013*; *Friston, 2005*; *Mumford, 1992*). Predictive coding computational models have implicated the prefrontal cortex (PFC) in higher-order predictions and PE processing (*Chennu et al., 2016*; *Garrido et al., 2008*; *Garrido et al., 2009a*; *Phillips et al., 2015*; *Phillips et al., 2016*). However, studies addressing the role of subregions of PFC in predictive processing are sparse. The dorsal part of medial PFC (dmPFC) is thought to be specialized for reporting error as a deviation from predicted events (*Alexander and Brown, 2011*). The ventral part of medial PFC (vmPFC), that is, OFC, has been proposed to integrate perceptual input from cortical and subcortical areas, together with memories of previous stimuli, to determine the current task context (*Wilson et al., 2014*). DmPFC and OFC subregions likely interact (*Alexander and Brown, 2014*), but the nature of the interplay and the specific role of the OFC remains to be delineated.

Numerous electrophysiological studies have used the predictive coding framework to explain event-related potentials (ERPs), including the mismatch negativity (MMN) and the P3 complex, as PE signals (*Chennu et al., 2016*; *Garrido et al., 2009a*; *Chennu et al., 2013*; *Doricchi et al., 2021*; *Garrido et al., 2007*; *Lieder et al., 2013*; *Wacongne et al., 2011*; *Hoy et al., 2021*). The fronto-centrally distributed MMN is usually elicited by an infrequent tone that differs in its acoustic properties (e.g., pitch, loudness) from a monotonous sequence of preceding frequent tones and persists in the absence of overt attention (*Näätänen et al., 1978*; *Näätänen et al., 2007*). Within the predictive coding framework, the MMN is interpreted as an early PE signal arising from prediction violation due to a top-down predictive contribution (*Garrido et al., 2009a*; *Garrido et al., 2007*; *Parmentier et al., 2011*; *Winkler, 2007*). In agreement with a hierarchical model transmitting predictions in a top-down

fashion to lower sensory areas are the findings of fMRI, scalp- and intracranial EEG studies showing MMN generators in the superior temporal planes bilaterally (i.e., primary auditory cortices and superior temporal gyri), in bilateral prefrontal cortices (i.e., inferior frontal gyri), as well as the inferior circular sulcus of the insula (*Garrido et al., 2009a*; *Phillips et al., 2015*; *Phillips et al., 2016*; *Opitz et al., 2002*; *Doeller et al., 2003*; *Liebenthal et al., 2003*; *Molholm et al., 2005*; *Shalgi and Deouell, 2007*; *Deouell, 2007*; *Garrido et al., 2009c*; *Nourski et al., 2018*; *Blenkmann et al., 2019*). The P3 complex (P3a and P3b), sensitive to the detection of unpredictable auditory events, is elicited after the MMN (*Chennu et al., 2013*; *Doricchi et al., 2021*; *Wacongne et al., 2011*; *El Karoui et al., 2015*; *Kompus et al., 2020*). The earlier and more fronto-centrally distributed P3a is evoked by infrequent or novel stimuli (*Polich, 2007*) and is typically interpreted as reflecting an involuntary attentional reorienting process (*Escera and Corral, 2007*). The P3b, with a centro-posterior maximum, is associated with context updating in WM (*Polich, 2007*; *Donchin and Coles, 1988*; *Chao et al., 1995*) and allocation of attentional resources to stimulus evaluation, (*Picton, 1992*) and is dependent on conscious awareness (*Sergent et al., 2005*; *Del Cul et al., 2007*; *Bekinschtein et al., 2009*).

Early PFC lesion studies using simple oddball tasks showed diminished MMN elicited by deviant tones within sequences of standard tones (*Alho et al., 1994*; *Alain et al., 1998*). In the same vein, PFC injury reduced the amplitude of the frontal P3a to unexpected novel auditory stimuli, whereas the parietally distributed P3b to detected deviants remained intact (*Knight, 1984*; *Knight and Scabini, 1998*; *Løvstad et al., 2012*). *Løvstad et al., 2012* reported attenuation of the P3a to unexpected novel environmental sounds after damage to either lateral PFC (LPFC) or OFC. Moreover, *Solbakk et al., 2021* showed that the N2b-P3a complex elicited by unexpected auditory sensory outcomes of self-initiated actions in OFC patients did not differ as clearly from the response to expected outcomes as for the healthy controls. Altogether, these findings indicate that OFC integrity might be necessary for the generation of the auditory MMN and P3a, and support the idea that the OFC is involved in predicting different types of perceptual events and outcomes, and in noticing violations of expectation.

Given the suggested involvement of OFC in predicting perceptual events and noticing violations of expectation, this study investigated whether the OFC is involved in the detection of PEs while processing auditory stimuli at two levels of abstraction. To this aim, we manipulated both the local and global auditory features of the environment using a novel variant of the oddball paradigm originally devised by *Bekinschtein et al., 2009*. The paradigm allows probing of hierarchical predictive processing by simultaneously violating expectations at two processing levels and timescales: A low 'local' level with a short timescale (i.e., tone-onset asynchrony: 150 ms) and a higher 'global' level with a longer timescale (i.e., sequence-onset asynchrony: seconds). At the local level, the regularity is determined by the transition probability (i.e., the probability with which a given stimulus follows another) *between tones within sequences*, while at the global level, the regularity is established by the transition probability *between sequences* as they unfold over a longer time frame. The factorial (2 × 2) structure of the paradigm enables measuring predictive processing at one level along with the feedforwarding of PEs to the next level, as well as the interaction between processing levels (i.e., PEs at both local and global levels). By dissociating neural responses elicited at hierarchical levels of acoustic regularity, the paradigm has provided evidence for hierarchical stages in auditory processing linked to the MMN and P3 complex in the healthy human brain (*Chennu et al., 2016*; *Chennu et al., 2013*; *Wacongne et al., 2011*; *Kompus et al., 2020*). We first extended knowledge about the electrophysiological modulations (i.e., ERPs) elicited by auditory deviance processing at the 'local' and 'global' levels by studying the interaction of these processing levels with the new 'local + global' level condition in a cohort of healthy adults. We then examined the impact of OFC injury on these ERP markers by comparing healthy adults to patients with OFC lesions. Based on previous findings in oddball tasks manipulating local levels only, we expected attenuated MMN amplitude following OFC lesions. Because the OFC has been implicated in generating predictions about perceptual events and actions, we also predicted dampened amplitudes of the P3 complex, which has been linked with deviance processing at a higher (i.e., 'global') hierarchical level.

## Results

Scalp EEG recordings were obtained during an auditory local–global paradigm from a control group of healthy adults, a group of patients with lesions to the OFC (*Figure 1*) and a lesion control group of patients with lesions to the LPFC (*Figure 1—figure supplements 2 and 3*). The experiment had

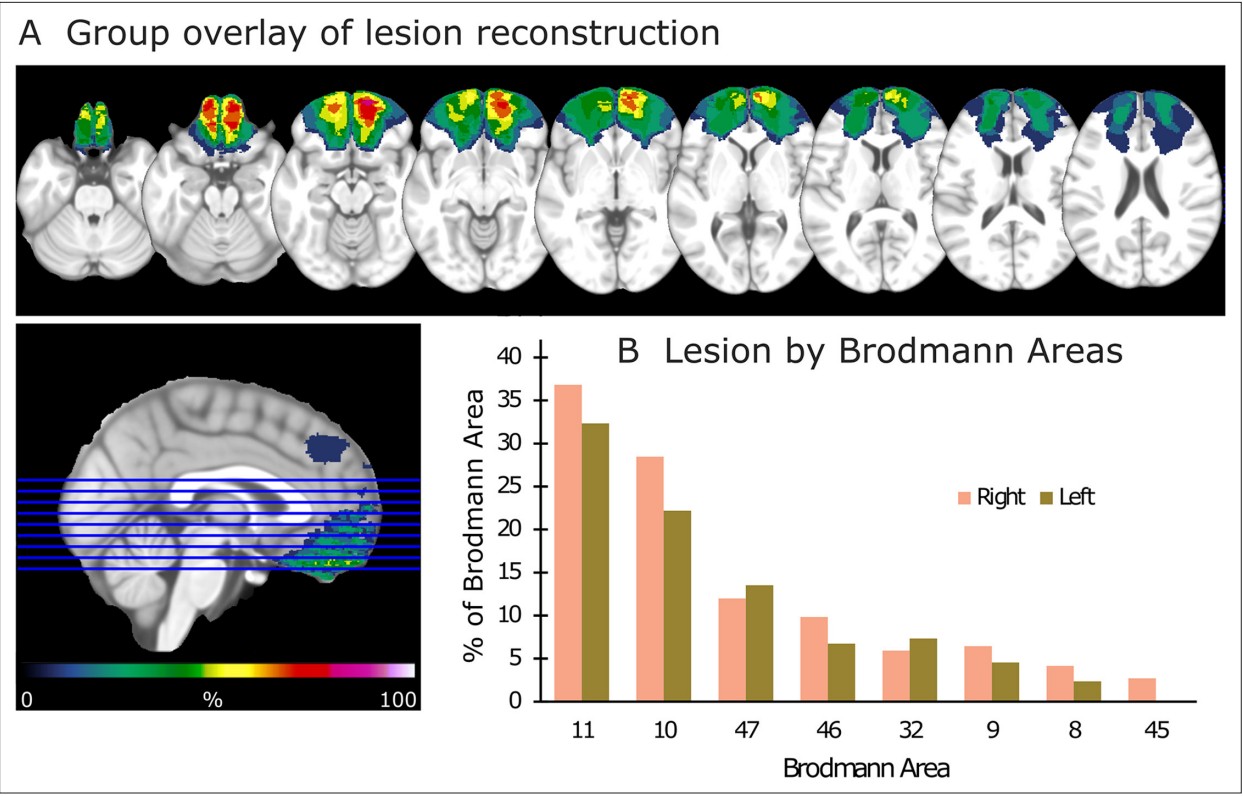

**Figure 1.** Lesion reconstruction for the group with orbitofrontal cortex (OFC) damage. (**A**) Aggregate lesion overlay maps in axial view. The color code (from 0 to 100%) for the group overlay indicates the percentage of shared lesion coverage across patients. The redder the color, the greater the lesion overlap. Neurological convention: the right side of the brain is depicted on the right side of the image and vice versa. (**B**) Average percentage of damaged tissue within each Brodmann area (BA) per hemisphere. BAs with less than 2% damage are not presented.

The online version of this article includes the following figure supplement(s) for figure 1:

**Figure supplement 1.** Lesion reconstructions for the orbitofrontal cortex (OFC) group.

**Figure supplement 2.** Lesion reconstructions for the right lateral prefrontal cortex (PFC) group.

**Figure supplement 3.** Lesion reconstructions for the left lateral prefrontal cortex (PFC) group.

**Figure supplement 4.** Lateral prefrontal cortex (PFC) lesion by Brodmann areas (BA).

regular and irregular blocks, where the most common sequence could be either regular (xxxxx) or irregular (xxxxy), building up global expectations (i.e., predictions). When the last tone in the sequence is different from the previous, it constitutes a local deviant. On rare occasions, deviant sequences were introduced, violating the global rule. Participants were asked to detect these global violations, which might simultaneously conform or not the local rules. In total, four key conditions were generated: control (predictable tones at both local and global levels), local deviant (local rule violations), global deviant (global rule violations), and local + global deviant (combined local and global rule violations). EEG-derived ERPs for the four conditions were analyzed and three condition contrasts were conducted to assess neuronal markers of deviance processing and isolate low- and high-level prediction violations: Control vs. local deviant, control vs. global deviant, and control vs. local + global deviant (see details in *Figure 2*). Behavioral data were collected through written reports of the number of rare sequences detected per task block. Amplitude and latency of time-locked responses to pairs of experimental conditions and group differences were compared using nonparametric cluster-based permutation tests and independent samples *t*-tests (see details in 'Materials and methods').

## Behavioral performance

Participants performed the task properly with an average error rate of 9.54% (SD 8.97) for the healthy control participants, 10.55% (SD 6.18) for the OFC lesion group, and 6.37% (SD 5.79) for the LPFC lesion group. There was no statistically significant difference between the counts of rare tone

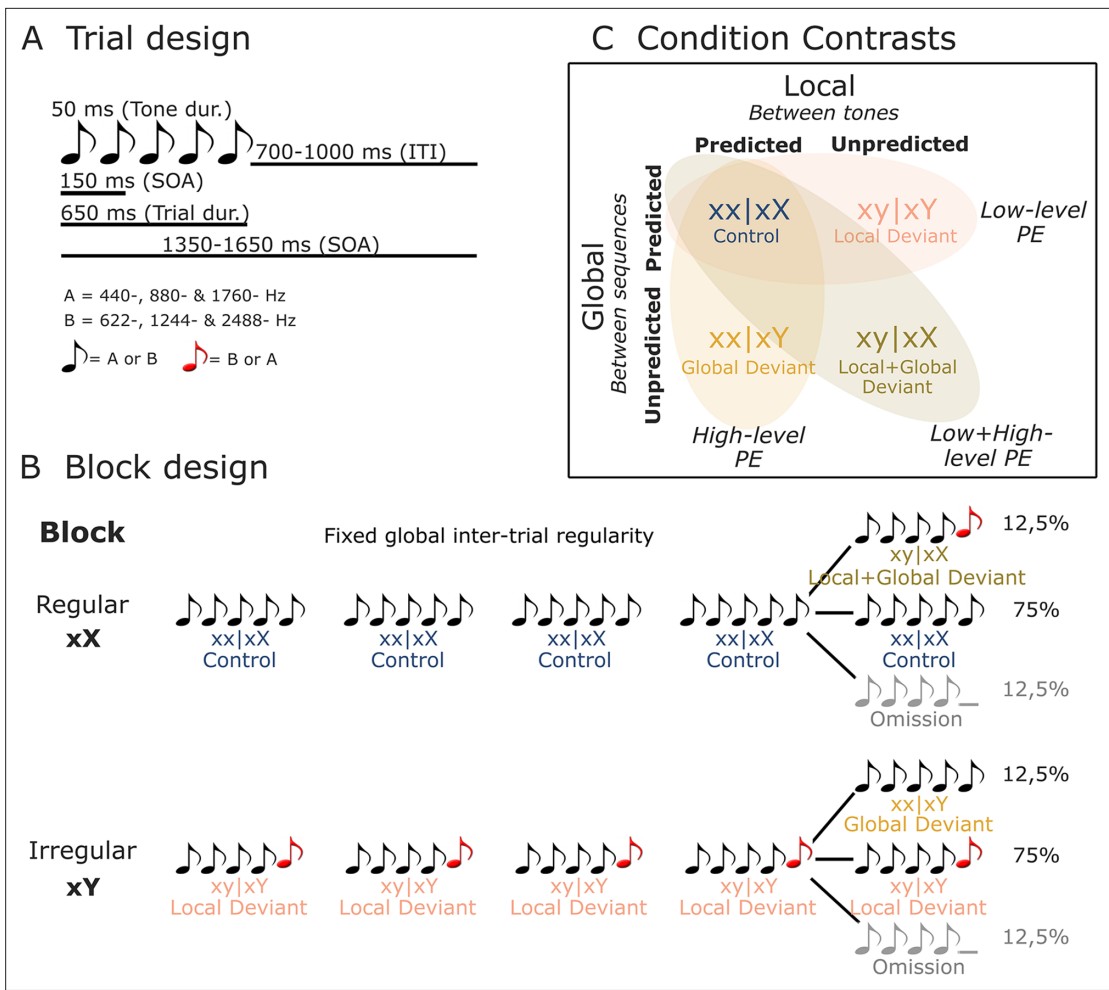

**Figure 2.** Illustration of stimuli and experimental design. (**A**) On each trial, five or four complex tones of 50 ms duration each were presented with a fixed SOA of 150 ms. Two types of tones were used to generate these trials: tone A (composed of 440, 880, and 1760 Hz sinusoidal tones) and tone B (composed of 622, 1244, and 2488 Hz sinusoidal tones). (**B**) Each block started with 20 frequent sequences of tones to establish the block's global rule. In regular xX blocks, standard sequences (75%) consisted of five repetitions of the same tone (i.e., xx|xX or control trials). These were interspersed with rare local deviant sequences (12.5% each) where the fifth sound was either different in frequency type (i.e., xy|xX or local + global deviant trials), or was omitted. The irregular xY blocks were similar, except that the standard sequences (75%) had a fifth sound differing in frequency type (i.e., xy|xY or local deviant trials), interspersed with rare (12.5%) local standard sequences (i.e., xx|xY or global deviant trials), or omission sequences. (**C**) By contrasting control (xx|xX) trials with local deviant (xy|xY), global deviant (xx|xY), and local + global deviant (xy|xX) trials, we isolated low-level, high-level, and combined low- and high-level prediction error (PE) responses, respectively. dur., duration; ITI, inter-trial interval; SOA, stimulus onset asynchrony.

sequences of the CTR group compared to the OFC group [$F_{(1, 24)}$ = 0.11, p=0.75], or the LPFC group [$F_{(1, 22)}$ = 0.96, p=0.34]. Participants from the CTR and OFC groups had a trend-level lower error rate in the irregular block (CTR: 8.39 ± 8.24%; OFC: 7.50 ± 7.34%) compared to regular block (CTR: 10.69 ± 11.36%; OFC: 13.60 ± 10.97%) [$F_{(1, 24)}$ = 3.55, p=0.07]. There was no block × group (CTR vs. OFC) interaction effect [$F_{(1, 24)}$ = 0.73, p=0.40]. This was not the case when contrasting the LPFC with the CTR group for the two blocks (LPFC; irregular block: 6.56 ± 7.73%; regular block: 6.18 ± 6.24%). There was no block [$F_{(1, 22)}$ = 0.31, p=0.58], or block × group interaction effect [$F_{(1, 22)}$ = 0.61, p=0.44].

## EEG results

### Local deviance response: MMN and P3a components

Analysis of the local deviance response revealed that ERPs to local-level unpredicted tones (xy|xY trials) differed significantly from local-level predicted tones (xx|xX trials). Both groups showed condition differences corresponding to a negative cluster in the data at 67–128 ms (i.e., MMN) for the CTR group [$t_{(13)}$ = –6633.65, p=0.012, 61/64 channels] and at 73–131 ms for the OFC group [$t_{(11)}$ =

–3734.49, p=0.035, 41/64 channels]. This was followed by a positive cluster at 143–313 ms (i.e., P3a) for the CTR group [$t$(13) = 24808.43, p<0.001, 60/64 channels], which extended from 145 to 344 ms for the OFC group [$t$(11) = 21796.75, p<0.001, 58/64 channels].

Testing for group differences was done in the time range of the MMN (i.e., 50–150 ms, based on the statistical analysis of the CTR and OFC group condition contrasts). Analysis of the response to local-level unpredicted tones (xy|xY trials) revealed a reduced MMN for the OFC patients compared to the CTR participants in a time window from 73 to 110 ms [positive cluster: $t$(25) = 1396.79, p=0.028, 39/64 channels]. Group differences in the condition difference waveforms (i.e., xy|xY minus xx|xX trials) yielded similar results [$t$(25) = –1090.62, p=0.02]. However, the groups did not differ significantly in the time range (i.e., 140–350 ms) of the P3a (positive cluster: p=0.22, negative cluster: p=0.53). Latency analysis for the MMN and P3a did not show statistically significant differences between the two groups. See *Figure 3* for a visual representation of the ERP waveforms of the CTR and OFC groups.

To better understand the nature of the ERP group differences revealed by the cluster-based permutation tests, complementary analysis on the mean amplitudes of the MMN and P3a components was conducted. The MMN was defined as the most negative peak in a post-stimulus window of 50–150 ms, and the P3a as the most positive peak in a post-stimulus window of 130–310 ms. The mean amplitude was calculated centered ±25 ms around individual peaks. The independent samples $t$-tests comparing the distinct components mean amplitudes between the two groups for the midline sensors revealed amplitude differences for the MMN [AFz (p=0.021), Fz (p=0.008), CPz (p=0.015), and Pz (p<0.001)] and for the P3a [AFz (p<0.001), Fz (p<0.001), FCz (p<0.001), and Cz (p=0.002)] (*Supplementary file 1a*\*).

Interestingly, responses to the standard tones predicted at both levels (xx|xX trials) did not diverge significantly between the OFC and the CTR group (positive cluster: p=0.79, negative cluster: p=0.51), or between the LPFC and the CTR group (positive cluster: p=1) (*Figure 3—figure supplement 1*). Analysis of the local deviance response in the time ranges of the MMN (i.e., 50–150 ms) and P3a (i.e., 140–350 ms) for the LPFC lesion control group did not reveal statistically significant differences between the LPFC and the CTR group (MMN, negative cluster: p=0.85; P3a, negative cluster: p=0.99) (*Figure 3—figure supplement 2*).

## Global deviance response: P3b component

We examined the presence of only a high-level deviance response, that is, global deviance, by comparing globally unpredicted tones (global deviant: xx|xY trials) with globally predicted ones (control: xx|xX trials), while keeping the local-level predictions fulfilled. Results revealed a significant condition effect captured by a positive cluster at posterior electrodes, which lasted from 381 to 714 ms for the CTR group [$t$(13) = 10521.05, p=0.009, 33/64 channels], and from 419 to 799 ms for the OFC group [$t$(11) = 13120.99, p=0.01, 51/64 channels]. This response was long-lasting with no well-defined peak and had the classical posterior maximum scalp topography and latency of the P3b (*Figure 4A*).

No statistically significant differences between the two groups were found in the time window (i.e., 380–800 ms, based on CTR and OFC group condition contrasts) of the P3b (positive cluster: p=0.61, no significant negative cluster was detected) (*Figure 4B*). Latency analysis for the P3b did not show statistically significant differences between the two groups. No significant differences were found between the LPFC and CTR groups in the time window of the P3b (i.e., 380–800 ms) (negative cluster: p=0.77) (*Figure 4—figure supplement 1*).

## Local + global deviance response: MMN, P3a, and P3b components

The analysis of the local + global deviance response resulting from the comparison of tones unpredicted at both local and global levels (local + global deviant: xy|xX trials) with tones predicted at both levels (control: xx|xX trials) revealed a condition effect for both groups. The CTR group showed a deviance response (i.e., two-peak and long-lasting MMN) induced by a deviant tone that is also unpredicted by the global rule, as indexed by a negative cluster with a frontal scalp distribution at 71–186 ms [$t$(13) = –7229.33, p=0.016, 59/64 channels]. Moreover, the OFC patients showed a similar MMN to unpredicted deviant tones at 67–139 ms, which did not reach significance when comparing with predicted standard tones [negative cluster: $t$(11) = –3690.11, p=0.068]. Following the MMN,

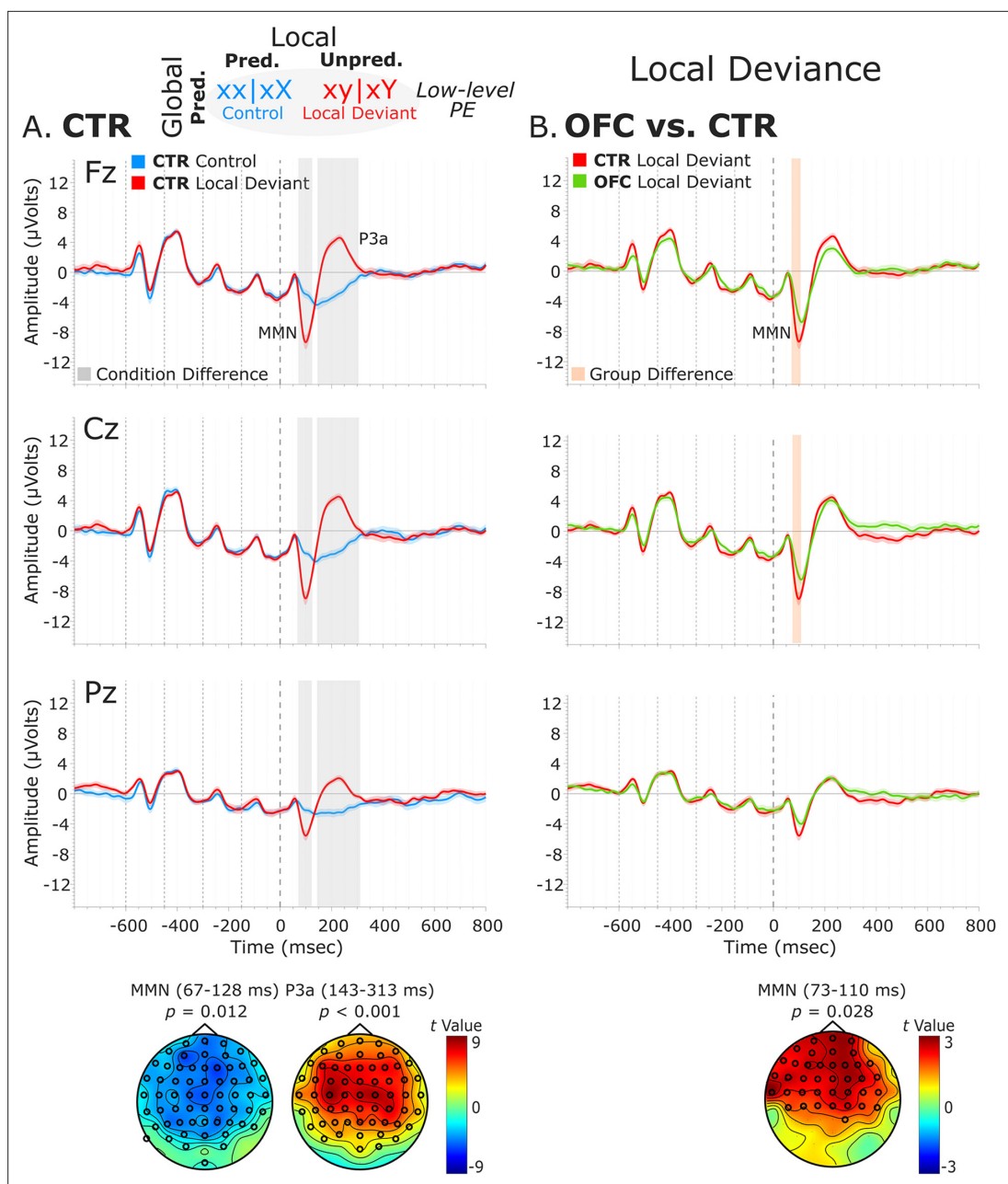

**Figure 3.** Local deviance event-related potentials (ERPs). (**A**) Local deviance response CTR group. To the left are the healthy control participants' grand average ERP waveforms at midline electrodes (from top to bottom: Fz, Cz, and Pz). ERPs from the processing of (standard) tones predicted at both levels (control: xx|xX trials) are in blue, and ERPs from the processing of (deviant) tones unpredicted at the local level (local deviant: xy|xY trials) are in red. Gray shaded bars indicate times when the electrode was part of a cluster showing significant within-group condition differences. At the bottom, the topographic scalp maps represent the statistical difference values for the *t*-contrast of the two experimental conditions computed for the time window corresponding to the cluster with significant differences. (**B**) Group differences (CTR vs. OFC). To the right are the CTR and OFC grand average ERP waveforms at the same midline electrodes. ERPs from the processing of tones unpredicted at the local level (local deviant: xy|xY trials) are in red for CTR and in green for OFC. Orange shaded bars indicate times when the electrode was part of a cluster showing significant differences between the groups. At the bottom, the topographic scalp map represents the statistical difference values for the *t*-contrast of the two groups computed for the time window corresponding to the cluster showing differences. MMN and P3a latencies did not show statistically significant differences between groups. Dashed lines at −600, −450, −300,−150, and 0 ms depict tone onsets. Shaded areas around the waveforms represent the standard error of the mean (SEM). OFC, orbitofrontal cortex.

The online version of this article includes the following figure supplement(s) for figure 3:

**Figure supplement 1.** Predicted standard tone response.

**Figure supplement 2.** Local deviance response. group differences (CTR vs. LPFC).

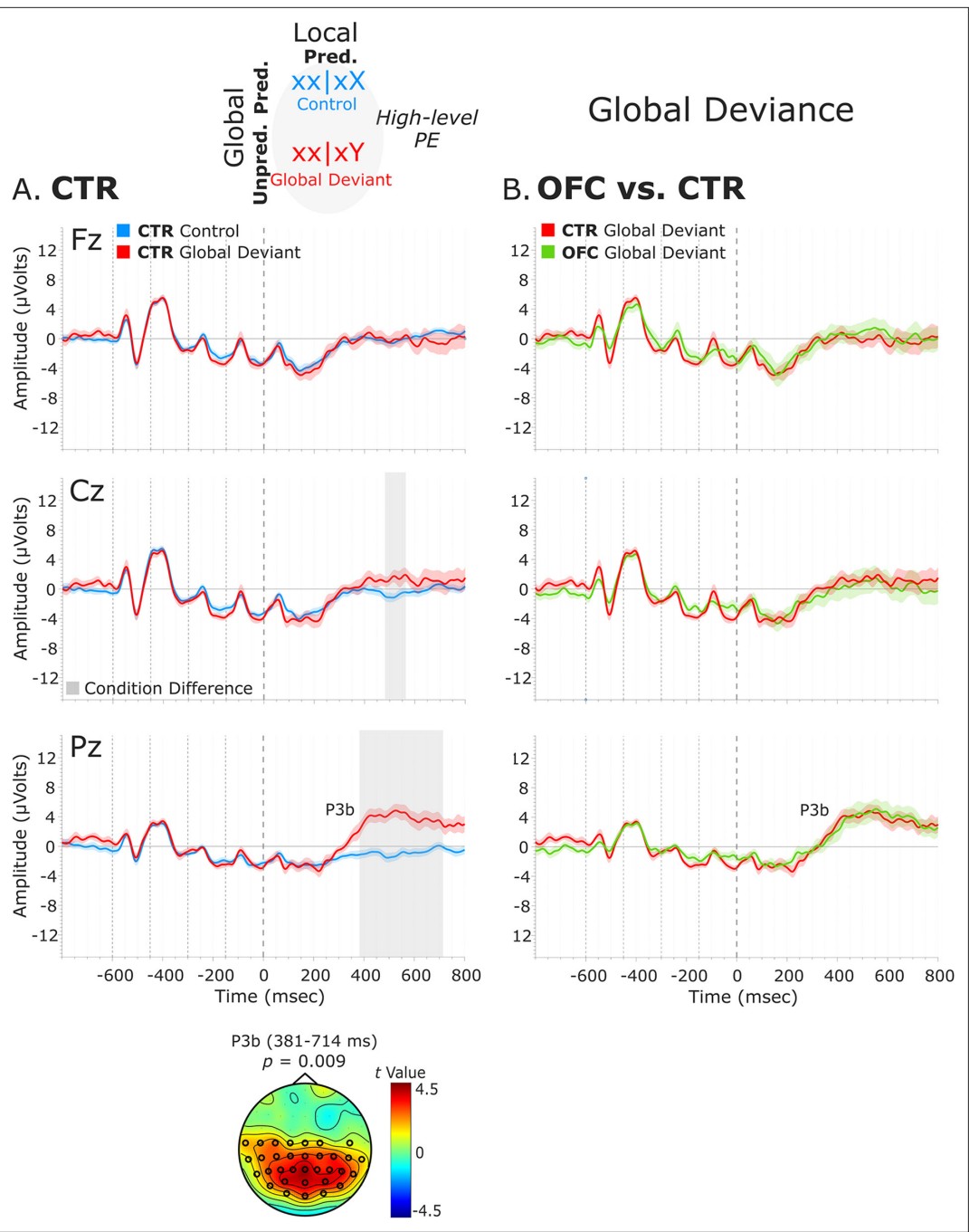

**Figure 4.** Global deviance event-related potentials (ERPs). (**A**) Global deviance response CTR group. To the left are the healthy control participants' grand average ERP waveforms at three midline electrodes (from top to bottom: Fz, Cz, and Pz). ERPs from the processing of (standard) tones predicted at both local and global levels (control: xx|xX trials) are in blue, and ERPs from the processing of (standard) tones unpredicted only at the global level (global deviant: xx|xY trials) are in red. Gray shaded bars indicate times when the electrode was part of a cluster showing significant within-group condition differences. At the bottom, the topographic scalp map represents the statistical difference values for the *t*-contrast of the two experimental conditions computed for the time window corresponding to the cluster showing significant differences. (**B**) Group differences (CTR vs. OFC). To the right are the CTR and OFC grand average ERP waveforms at the same midline electrodes. ERPs from the processing of standard tones unpredicted at the global level (global deviant: xx|xY trials) are in red for the CTR group and green for the OFC group. P3b latency did not show statistically significant differences between groups. Dashed lines at

*Figure 4 continued on next page*

*Figure 4 continued*

−600, −450, −300, −150, and 0 ms depict tone onsets. Shaded areas around the waveforms represent the standard error of the mean (SEM). OFC, orbitofrontal cortex.

The online version of this article includes the following figure supplement(s) for figure 4:

**Figure supplement 1.** Global deviance response. group differences (CTR vs. LPFC).

a P3 complex (composed of a frontally distributed P3a and a posteriorly distributed P3b response) was observed in CTRs, as indexed by a positive cluster extending from around 188–684 ms [$t(13)$ = 42019.65, p<0.001, 63/64 channels]. A similar P3 complex was found for the OFC patients as indexed by a positive cluster from 217 to 710 ms [$t(11)$ = 33590.81, p<0.001, 64/64 channels].

Group-level statistics performed on the MMN time window (i.e., 70–185 ms, based on CTR and OFC group condition contrasts) showed a trend-level reduced MMN for OFC patients compared to controls in a time window between 81 and 106 ms [positive cluster: $t(25)$ = 588.31, p=0.066, 34/64 channels]. Group differences in the difference waveforms (i.e., xy|xX minus xx|xX trials) yielded similar results [$t(25)$ = −314.64, p=0.068]. Latency analysis showed that the MMN was not significantly delayed at midline electrodes [only a trend-level effect at FCz: $t(24)$ = 2.26, p=0.033; and Cz: $t(24)$ = 1.78, p=0.088 electrodes, which did not survive the false discovery rate (FDR) correction]. Group-level statistics in the time window of P3a (i.e., 180–485 ms) revealed an attenuated P3a for OFC patients compared to CTR participants in a time window from 184 to 274 ms [negative cluster: $t(25)$ = −3730.34, p=0.024, 59/64 channels], and from 329 to 479 ms [positive cluster: $t(25)$ = 3063.91, p=0.03, 36/64 channels]. Latency analysis showed that the P3a emerged 60.82 ms later, on average, in the OFC compared to the CTR group. The latency difference was significant for all the midline electrodes [Fz: $t(24)$ = 4.91, p<0.001; FCz: $t(24)$ = 4.64, p<0.001; Cz: $t(24)$ = 4.75, p<0.001; CPz: $t(24)$ = 3.46, p=0.002; Pz: $t(24)$ = 4.34, p<0.001]. *Table 1* shows the 50% area latencies for MMN and P3a. No statistically significant differences between the two groups were found in the time window of the P3b (i.e., 450–710 ms) for both amplitude and latency analysis. *Figure 5* illustrates these results.

To provide clarity regarding whether the MMN and P3a group differences revealed by the cluster-based permutation tests are amplitude differences or outcomes of latency variations, complementary analysis of the mean amplitudes of the MMN and P3a components was conducted. The MMN was defined as the most negative peak in a post-stimulus window of 50–250 ms and the P3a as the most positive peak in a post-stimulus window of 150–350 ms. The mean amplitude was calculated centered ±40 ms around individual peaks. The independent samples $t$-tests comparing the distinct components mean amplitudes between the two groups for the midline sensors revealed amplitude differences for the MMN [AFz (p=0.007), FCz (p=0.051), Cz (p=0.004), CPz (p=0.002), and Pz (p<0.001)], but not for the P3a (*Supplementary file 1b*). Thus, the group differences for the P3a elicited by the local + global deviance seem to be a by-product of latency differences.

Group-level statistics performed on the MMN, P3a, and P3b time windows did not show significant differences between LPFC patients and controls (MMN, positive cluster: p=0.74; P3a, positive cluster: p=0.52; P3b, positive cluster: p=0.46). See *Figure 5—figure supplement 1*.

A complementary analysis comparing the processing of local deviant with local + global deviant in the CTR group showed significant differences in both ERPs' amplitude, negative cluster at 85–210 ms [$t(13)$ = −8021.49, p=0.018] followed by positive cluster at 221–684 ms [$t(13)$ = 35780.80, p<0.001] (*Figure 5—figure supplement 2A*), and latency (*Supplementary file 1c*). The OFC group analysis yielded similar results: a negative cluster at 85–231 ms [$t(11)$ = −7984.18, p=0.014] succeeded by a positive cluster at 245–680 ms [$t(11)$ = 18643.85, p<0.001]. Group-level statistics performed on the difference waveforms (i.e., local + global minus local deviant) revealed significant or trend-level differences in amplitude [positive cluster at 131–161 ms ($t(25)$ = 466.94, p=0.066), negative cluster at 186–253 ms ($t(25)$ = −2741.61, p=0.018), and positive cluster at 319–461 ms ($t(25)$ = 1695.83, p=0.054)] (*Figure 5—figure supplement 2B*), and significant differences in latency (*Supplementary file 1d*).

In summary, in both groups, local-level deviant tones produced an early MMN followed by a frontally distributed P3a. Local-level standard tones, which were unpredicted by the global rule, elicited only a posterior P3b. Local-level deviant tones in sequences unpredicted by the global rule produced a long-lasting MMN, followed by a P3 complex (frontal P3a and posterior P3b). In OFC patients, ERPs elicited by local-level standard tones, both predicted and unpredicted by the global rule, were

**Table 1.** 50% area latency of MMN and P3a for the local + global deviance response.

| | MMN | | | | P3a | | | |
|---|---|---|---|---|---|---|---|---|
| | Latency (SD) | | Diff. (ms) | p-Value | Latency (SD) | | Diff. (ms) | p-Value |
| | CTR | OFC | | | CTR | OFC | | |
| Fz | 127.40 (20.09) | 138.28 (26.14) | 10.88 | 0.242 | 279.88 (21.37) | 334.90 (35.13) | 55.01 | <0.001 |
| FCz | 127.54 (20.07) | 148.21 (26.54) | 20.67 | 0.033 | 280.02 (26.27) | 332.78 (31.74) | 52.76 | <0.001 |
| Cz | 128.38 (19.57) | 147.07 (33.21) | 18.69 | 0.088 | 282.95 (28.77) | 352.47 (45.17) | 69.52 | <0.001 |
| CPz | 129.07 (23.87) | 139.67 (30.96) | 10.89 | 0.330 | 283.79 (24.57) | 347.59 (63.79) | 63.80 | 0.002 |
| Pz | 130.61 (22.36) | 149.19 (33.54) | 18.58 | 0.105 | 292.58 (33.90) | 355.57 (40.10) | 62.99 | <0.001 |

50% area latency measures in milliseconds (ms) from the onset of the fifth tone of the sequence for the MMN and the P3a components, separately for the healthy control participants (CTR) and the OFC lesion patients (OFC). Diff. is the latency difference between the two groups (OFC vs. CTR) given in ms; p-values as a result of independent samples t-tests comparing the component's 50% area latency between the two groups. Standard deviation (SD) is given in brackets.
OFC, orbitofrontal cortex; MMN, mismatch negativity.

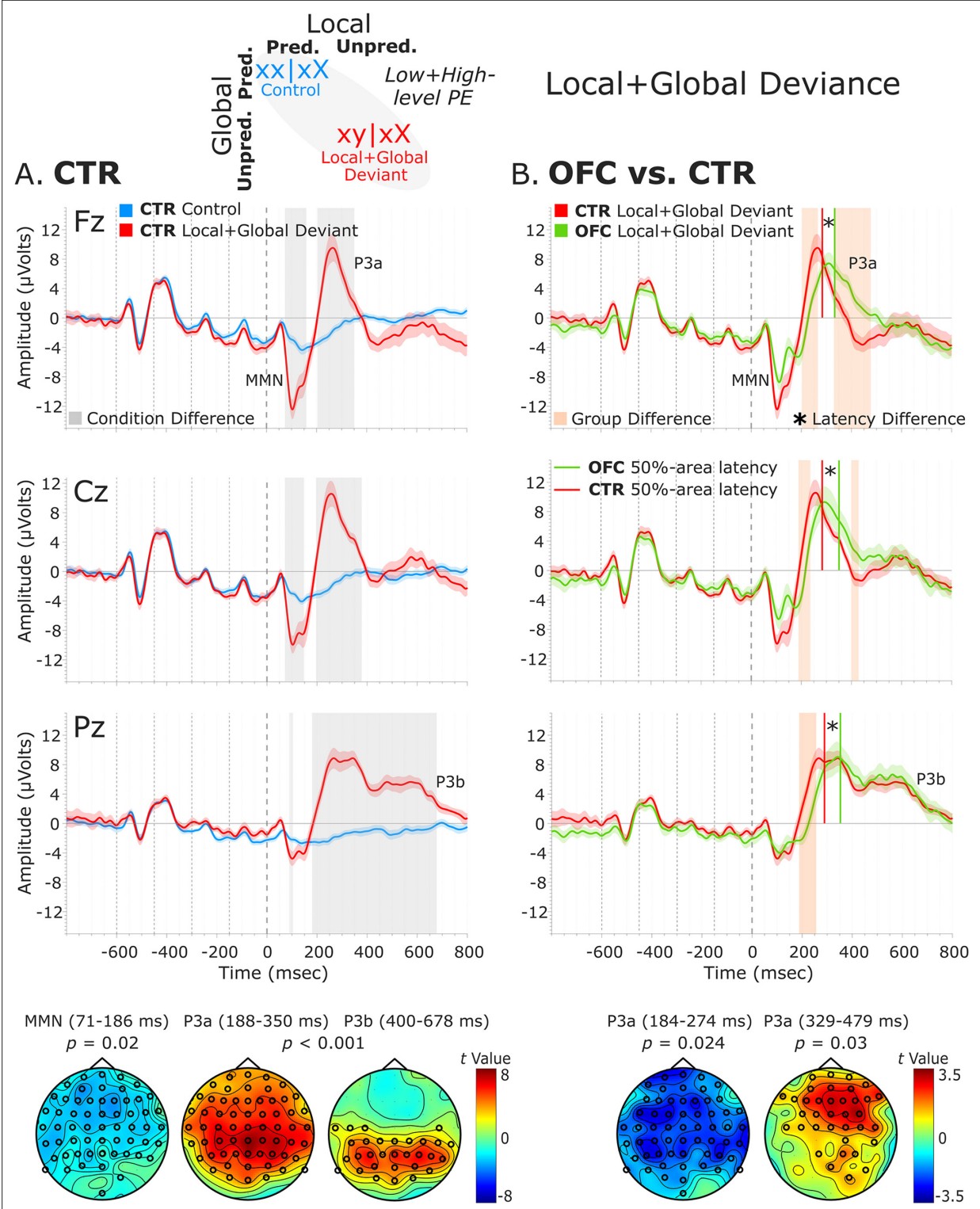

**Figure 5.** Local + global deviance event-related potentials (ERPs). (**A**) Local + global deviance response CTR group. To the left are the healthy control participants' grand average ERP waveforms at midline electrodes (from top to bottom: Fz, Cz, and Pz). ERPs from the processing of (standard) tones predicted at both local and global levels (control: xx|xX trials) are in blue, and ERPs from the processing of (deviant) tones unpredicted at both levels (local + global deviant: xy|xX trials) are in red. Gray shaded bars indicate times when the electrode was part of a cluster showing significant within-group condition differences. At the bottom, the topographic scalp maps reflect the statistical difference values for the *t*-contrast of the two experimental conditions computed for the time window corresponding to the cluster with significant differences. (**B**) Group differences (CTR vs. OFC). To the right are

*Figure 5 continued on next page*

*Figure 5 continued*

the CTR and OFC grand average ERP waveforms at the same midline electrodes. ERPs from the processing of (deviant) tones unpredicted at both local and global level (local + global deviant: xy|xX trials) are in red for CTR and in green for OFC. Orange shaded bars indicate times when the electrode was part of a cluster showing significant differences between the groups. Vertical red lines indicate the 50% area latency for the CTR group while vertical green lines indicate the latency for the OFC group for the corresponding components (i.e., MMN and P3a). Asterisks (*) denote significant latency differences. At the bottom, the topographic scalp maps represent the statistical difference values for the *t*-contrast of the two groups computed for the time window corresponding to the cluster with observed differences. Dashed lines at −600, −450, −300, −150, and 0 ms depict tone onsets. Shaded areas around the waveforms represent the standard error of the mean (SEM). OFC,orbitofrontal cortex; MMN, mismatch negativity.

The online version of this article includes the following figure supplement(s) for figure 5:

**Figure supplement 1.** Local + global deviance response. group differences (CTR vs. LPFC).

**Figure supplement 2.** Local versus local + global deviance event-related potentials (ERPs).

unaffected compared to healthy controls. However, patients showed an attenuated MMN to local-level deviant tones, as well as attenuated MMN and delayed P3a to local + global-level deviant tones, while the P3b was unaffected. The LPFC group did not differ from the healthy controls on any ERP measures.

## Discussion

We aimed to delineate the involvement of the OFC in detecting violations of predictions (i.e., PE) at the two hierarchical levels of the local–global paradigm. To this end, we studied the neurophysiological markers of auditory deviance processing in patients with focal OFC lesions and healthy controls while occasionally violating their predictions at a local (i.e., *between tones within sequences*) and at a global (i.e., *between sequences*) timescale. By attending to tone sequences and counting any rare sequences, we expected that participants would extract the global rule that characterized deviant sequences. The reported counts suggest that patients accomplished the counting properly despite their brain damage. Moreover, their general intellectual ability and scores on standardized neuropsychological tests did not differ significantly from the control group and were well within the normal range compared to normative data. This indicates that they did not have deficits in the types of learning, memory, psychomotor speed, and cognitive executive functioning tested (see Table 3). However, the ERP findings showed that processing of auditory stimuli that violated predictions at the local level was affected by the OFC lesion. OFC lesions also impacted processing of stimuli that violated predictions at both the local and global levels. On the other hand, processing of predicted stimuli at the local level (i.e., standard tones), even when these were unpredicted by the global rule, was preserved. Comparable effects were absent in patients with lesions restricted to the LPFC, which lends a degree of anatomical specificity to the altered predictive processing resulting from OFC lesions. These findings indicate that the OFC plays a role in the detection of local and local + global auditory PEs, thus providing a novel perspective on the involvement of this region in predictive processing.

### OFC lesions affect the processing of local-level deviants

At the local level (xy|xY), deviant auditory stimuli generated a low-level PE response indexed by two successive ERP components within an early 70–310 ms window, an MMN followed by a P3a. The ERP responses were comparable to those reported in previous studies using a similar experimental design (*Chennu et al., 2016*; *Chennu et al., 2013*; *Doricchi et al., 2021*; *Wacongne et al., 2011*; *Bekinschtein et al., 2009*). These studies have traditionally interpreted the short-term (i.e., local level) MMN as a bottom-up PE signal, which indexes the amount of information in each deviant event that is not explained away by top-down prediction signals. The P3a, which is sensitive to the local status of the deviant tone, has also been reported in previous scalp- (*Chennu et al., 2016*, *Doricchi et al., 2021*, *Kompus et al., 2020*, *Bekinschtein et al., 2009*) and intracranial (*El Karoui et al., 2015*) EEG studies. It has been associated with the orienting of attention towards the unexpected local deviant stimulus, or the evaluation of its contextual novelty, that is, how novel the tone is when considered in a global context (*Escera and Corral, 2007*). The OFC patients had ERPs with similar morphology and scalp distribution as the controls, but the MMN amplitude was significantly attenuated. This dampening of the MMN elicited by local-level deviants shows for the first time that the OFC is involved in low-level sensory processing of prediction violation.

## OFC lesions affect the processing of local- and global-level deviants

Most studies using the local–global paradigm tested the global violation by pooling over all rare sequences of tones (i.e., xx | xY and xy | xX trials), which, as per our assumptions, contains two different types of PE signals. We split this classical global condition into local + global and global deviance (see 'Materials and methods'). The former requires detection of a deviant following a sequence of four identical tones (i.e., low-level PE) and concurrently identifying this rare sequence as a global deviant (i.e., high-level PE). The latter requires detection of a rare sequence of five identical tones as a global deviant (i.e., only high-level PE). The detection of local + global deviance evoked patterns of brain activity comprised of a 'two-peak' and long-lasting MMN (70–185 ms) followed by a sustained positive polarity ERP, consisting of a frontally distributed P3a and a posteriorly distributed P3b response. The 'two-peak' appearance of the MMN, which was not present in response to the local deviance, may suggest overlapping of an 'early MMN' with a succeeding N2b component. The N2b has been associated with voluntary attention (*Näätänen et al., 1978*), contrary to the 'early MMN,' which is thought to reflect automatic and short-term processing of novelty (*Pegado et al., 2010*). In our study, the 'early MMN' would reflect accumulation of evidence based on short-lived echoic memory representation or, in predictive coding terms, an automatic, low-level PE response. On the other hand, the N2b, along with P3a, may indicate a stronger influence of voluntary attentional resources engaged in the local + global deviance processing (i.e., low- and high-level PE), where an accumulation of evidence on longer timescales is required.

Previous research showed that auditory sequences containing infrequent pitch deviations elicited both the MMN and P3 when participants were instructed to focus their attention on the local rule (i.e., deviant tones). However, when participants were asked to attend to the global rule (i.e., the deviant sequences as a whole), the MMN was no longer elicited, while the P3 was maintained (*Sussman, 2007*). In the current study, participants were instructed to attend to the global rule, but the MMN was still evoked, and interestingly both the MMN and the P3a exhibited larger amplitudes and longer durations compared to those elicited by a local level violation (see *Figure 5—figure supplement 2*). In predictive coding terms, this response can be explained by positing a high-level prediction partially explaining away the low-level PE for the local level violation. This effect is consistent with previous studies (*Wacongne et al., 2011*; *Kompus et al., 2020*). indicating that violation of not only local but also global regularity is reflected in the MMN response. However, it stands in contrast to earlier studies suggesting that high-level violations are indexed uniquely by P3 responses (*Chennu et al., 2013*; *Wacongne et al., 2011*; *Bekinschtein et al., 2009*; *Cornella et al., 2012*; *Strauss et al., 2015*). This finding refines earlier results by showing that local and global effects are not entirely independent but rather interact within an early time window. Consequently, low- and high-level PEs are dependent on- and interact with each other, providing support not only for a hierarchical model, but also for a predictive rather than a feedforward one.

The OFC patients showed the same pattern of ERP responses as the healthy controls, but the MMN amplitude was attenuated, while the latency of P3a was prolonged by ≈ 60 ms. Consequently, the OFC lesions affect not only the neural responses elicited by the local-level violation, but also the responses elicited by violations which are unpredicted at the global level, and therefore require integration of information over longer timescales and at a higher level of abstraction. To our knowledge, only one previous study has reported reduced P3a amplitudes in OFC patients performing an auditory novelty oddball task (*Løvstad et al., 2012*) and no studies have reported latency effects. A larger number of studies have found reduced MMN to local-level auditory deviants after damage to the dlPFC, (*Alho et al., 1994*; *Alain et al., 1998*) while P3 reduction has been found in LPFC patients, indicating a deficit in the orienting response to unexpected and novel stimuli (*Knight, 1984*; *Daffner et al., 2000*; *Daffner et al., 2003*).

## Preserved processing of global-level deviants

The detection of global deviance elicited a late posterior P3b and no earlier ERP responses. Another study using the same paradigm demonstrated only P3b-like responses elicited by sequences of five identical tones unpredicted by the global rule (*Wacongne et al., 2011*). A sequence of five identical tones eliciting a novelty signal when participants expected a different sequence was seen by the authors as suggestive that the brain operates as a multilevel predictive system with P3b reflecting a high-level PE. Although we expected that OFC lesions would affect the P3b as well, our results

showed unaffected P3b responses for both global and local + global deviance detection. One possible explanation is that P3b is elicited by target detection and not deviance detection per se, that is, specifically linked to the detection of events that are salient or important to the current goal and memory-related processes (*Polich, 2007*; *Polich and Criado, 2006*; *Walentowska et al., 2016*). Note that as participants were instructed to count the total number of global deviations, these violations (i.e., xx|xY and xy|xX) were also the targets in the present study. Indeed, in oddball tasks, targets, but not other deviants, elicit large amplitude P3b responses associated with voluntary detection of infrequent and task-relevant stimuli (*Soltani and Knight, 2000*). Interestingly, when the same paradigm was used in patients suffering from disorders of consciousness, the P3 response to global novelty was only observed in patients who showed signs of consciousness (*Bekinschtein et al., 2009*; *Faugeras et al., 2012*; *Faugeras et al., 2011*). The P3b response in this task could thus reflect downstream processes related to conscious access per se (*Sergent and Naccache, 2012*; *Aru et al., 2012*) and not a high-level deviance detection signal. Moreover, lesions in the lPFC (*Knight, 1984*; *Knight and Scabini, 1998*; *Løvstad et al., 2012*) and the OFC (*Løvstad et al., 2012*) have been found to reduce the amplitude of the P3a but not the P3b, supporting normal target detection and suggesting that the OFC is not critical for P3b generation. Besides P3 potentials, early PE responses (60–220 ms) to global unexpected sequences have been observed using sequences of tones in scale (*Volehaugen, 2021*). Here, the response to global deviance (i.e., sequence of repetitive tones) is attenuated, and therefore, early effects of top-down prediction modulations might not be observable.

## Lack of findings in the LPFC lesion group

Intracranial studies examining local- and global-level PE detection have pointed to the role of the inferior frontal gyrus (IFG) as a frontal source supporting top-down predictions in MMN generation (*Phillips et al., 2016*; *Nourski et al., 2018*; *Dürschmid et al., 2016*; *Rosburg et al., 2005*). Responses to global deviants but not local deviants (i.e., no MMN) have been observed in the LPFC, but not the IFG (*El Karoui et al., 2015*). Additionally, studies employing dynamic causal modeling of MMN have frequently modeled frontal sources encompassing the IFG (*Garrido et al., 2009a*; *Phillips et al., 2015*). A review study highlighted the potential contributions of both IFG and middle frontal gyrus to MMN generation, suggesting that the specific source might vary depending on deviant characteristics, such as pitch or duration (*Deouell, 2007*). Based on these findings, which implicate the LPFC in predictive processing and the generation of MMN, we expected to find altered neural responses following LPFC lesions. In an early LPFC lesion study (*Alho et al., 1994*), diminished MMN to local-level deviants was reported, with the lesion cohort exhibiting a hemisphere ratio of 7/3 for left and right hemispheres, which is different from our cohort's ratio of 4/6. Furthermore, all individuals in that study had infarcts in the middle cerebral artery, resulting in a more uniform lesion location compared to our cohort. Notably, the lesions observed in our LPFC group appeared to be situated in more superior brain regions and toward the MFG compared to the predominantly reported involvement of the IFG in previous studies. Another factor that might contribute to no effects is the heterogeneity of the lesions in our LPFC group (see *Figure 1—figure supplements 1–4*). Especially for the left hemisphere cohort, the individual lesions did not share a consistent anatomical location. The right hemisphere cohort showed a higher degree of lesion overlap, but overall, the lesions were not centered in the IFG area, with the highest overlap being in the MFG area. This variation in lesion location could potentially explain the lack of effects observed in the present study.

## OFC and hierarchical predictive processing

In the context of predictive coding, the MMN is explained in terms of perceptual learning under hierarchical generative models of auditory input (*Friston, 2005*; *Baldeweg, 2006*). MMN is viewed as a cortical driven PE signal, which can only be accounted for by postulating a top-down predictive contribution (*Friston, 2005*; *Garrido et al., 2009a*; *Garrido et al., 2007*; *Parmentier et al., 2011*; *Winkler, 2007*). Numerous studies have identified MMN generators in both the superior temporal planes and the PFC, (*Opitz et al., 2002*; *Doeller et al., 2003*; *Liebenthal et al., 2003*; *Molholm et al., 2005*; *Shalgi and Deouell, 2007*; *Deouell, 2007*; *Jemel et al., 2002*), suggesting that it emerges from the deviant-induced suspension of neural adaptation within the primary auditory cortices, coupled with changes in temporo-frontal connections (*Garrido et al., 2008*; *Garrido et al., 2009a*; *Phillips et al., 2015*; *Phillips et al., 2016*; *Garrido et al., 2009c*). This perspective on the MMN supports the

existence of a hierarchy of internal models, wherein predictions are transmitted in a top-down manner to lower sensory areas. Within this framework, the observed changes in MMN responses following OFC damage show the role of this brain region in the neural circuitry underlying MMN generation and its contribution to the top-down process that modulates the deviance detection system in lower sensory areas. Specifically, the reduction in MMN amplitude in response to local prediction violations implicates a lesion-induced effect on sensory predictive signaling, possibly stemming from weaker formation of top-down predictions at the local level, that is, the weaker the prediction the lesser the mismatch reflected in the MMN response. Furthermore, the OFC lesion-related modulations extended to the processing of global regularities, as evidenced by the reduced MMN response accompanied by the delayed P3a response to violations involving both local and global predictions. The influence on ERPs indexing detection and attention to global-level violations might reflect a disturbance in the recurrent interactions between cortical regions (i.e., temporo-frontal connections), leading to weakened predictions in OFC patients. It could possibly reflect a reduced connection between PEs at lower hierarchical areas and predictions at higher areas.

An alternative interpretation is that the MMN solely reflects a passive, bottom-up process of adaptation to repeated stimuli (i.e., stimulus-specific adaptation; SSA), and this adaptation, rather than predictive processing, could be altered in OFC patients (*Auksztulewicz and Friston, 2016*; *Todorovic and de Lange, 2012*; *Carbajal and Malmierca, 2018*; *Garrido et al., 2009b*). How could adaptation and predictive accounts be distinguished? Findings from animal models indicate that both SSA and PE signals contribute to the generation of MMN with responses to deviant (non-repeated) and standard (repeated) tones reflecting active predictive activity and not simply SSA in single neurons. Specifically, this predictive activity followed a hierarchical pattern that extended from subcortical structures to the auditory cortex (*Parras et al., 2017*). However, while mismatch responses in the auditory areas were mainly induced by stimulus-dependent effects (e.g., SSA), auditory responsiveness in the PFC was driven by unpredictability, yielding context-dependent, comparatively delayed, more robust and longer-lasting mismatch responses mostly consisting of PE signaling activity (*Casado-Román et al., 2020*). Taking into consideration the anatomical distribution of SSA effects and the fact that in the present study the ERPs to standard tones were unaffected after OFC lesion, we suggest that SSA is intact and a specific disturbance in predictive processing is present in OFC patients. Note that OFC patients had a diminished MMN, but not an absolute absence of it, also pointing to a preserved SSA mechanism.

Temporal information derived from scalp- and intracranial ERP data provides valuable insights into the nature of the P3a and its role in predictive processing. P3a, typically considered a late stage of novelty processing, is intricately linked to the evaluation of the involuntary orienting response (*Escera et al., 2000*; *Escera et al., 2001*; *Friedman et al., 2001*). The component is also elicited when the novel stimulus is potentially task-relevant (*Schomaker and Meeter, 2014a*) and strongly dependent on the context in which it is presented (*Schomaker et al., 2014b*). Interestingly, the P3a has been source localized to the same anterior cingulate and PFC network that is involved in error processing (*Wessel et al., 2012*; *Wessel et al., 2014*). This is indicative that responses to novelty actually reflect PEs, which may result in a brain response that generates the P3a (*Schomaker et al., 2014c*). In terms of hierarchical predictive processing, the MMN traditionally signifies the local short-term deviance in a sequence of stimuli and the release of a PE signal. This signal contributes to the formation of global long-term predictions, which deal with the detection of local deviants across the entire pool of sequences, with the P3a indexing the PE signal. In the present study, the P3a was elicited in response to local regularity violation, where low-level perceptual expectation was violated, and in response to combined local and global regularity violation, where both perceptual and high-level conceptual expectations were violated, with the former being larger in amplitude and lasting longer. The more pronounced and later P3a response elicited by the joint violation of both expectations reflects a more global and integrative predictive system, which appears to be organized in several stages (*Winkler et al., 2005*) The observed delay in the P3a following OFC damage indicates the OFC's role in a later stage of predictive processing, which is based on global long-term predictions, thereby giving rise to high-level PEs. The latency of the P3 wave usually increases with increases in perceptual processing demands (*Johnson, 1986*) and yields a discrete measure of cognitive processing speed (*Duncan-Johnson and Kopell, 1981*). Moreover, latency change is generally considered a more reliable indicator of disturbed cognitive functioning than amplitude change, the former being more difficult to

modify with changes in attention (*Picton, 1992*; *Picton et al., 1992*). Therefore, we posit that the alteration of P3a may result from a delay in detecting and processing violations of prediction (i.e., PE) at a higher level of abstraction. Overall, the alterations observed in the electrophysiological signatures of PE signals (i.e., MMN and P3a) following OFC lesions suggest that the OFC supports a mechanism that maintains an internal representational model (i.e., a predictive code) of the external environment with the final aim of predicting events.

Although experimental evidence, reinterpreted with predictive coding, suggests PFC contribution to a top-down modulation of auditory deviance detection in primary auditory cortices (*Opitz et al., 2002*; *Doeller et al., 2003*), the role of the OFC in this process is rarely studied in humans. Our EEG findings are consistent with an intracranial EEG study in monkeys, (*Chao et al., 2018*), which reported the encoding of high-level PE and prediction update signals within the frontopolar PFC (BA 10) and the dlPFC. The ventrolateral PFC (BAs 44, 45, 47) is also implicated in the processing of complex auditory sequences (*Wilson et al., 2017*) and sequence-order PEs (*Wilson et al., 2015*), with studies employing the local–global paradigm showing high-level PEs confined to the ventrolateral PFC (*Uhrig et al., 2014*; *Wang et al., 2015*). Interestingly, some of the same brain areas (i.e., BAs 10, 47, 45) were damaged in our OFC cohort. Moreover, PET imaging revealed increased activity modulation in the OFC (i.e., BAs 11 and 13) in response to stimuli that deviated from expectation (*Petrides et al., 2002*) while a single-neuron study revealed selective responsiveness of neurons in BA 11 of macaques to novel but not familiar stimuli (*Rolls et al., 2005*). BA 11 within the OFC is proposed to capture novel information relative to current experience and expectations and to integrate this information to the higher cognitive processing occurring in the LPFC (*Petrides, 2007*).

The OFC is unique among areas in the PFC, connecting with all five sensory modalities and relevant memory and decision-making areas such as the hippocampus and striatum (*Du et al., 2020*). It might therefore be in a privileged position to generate predictions based on contextual and temporal structure in the environment, allowing quick adaptation to new rules (*Stalnaker et al., 2015*; *Wilson et al., 2014*). Moreover, studies on time perception (*Solbakk et al., 2021*; *Berlin et al., 2004*) support the importance of an intact OFC in maintaining temporal information needed to sustain a stable map of task context for longer periods, ultimately optimizing predictions (*Schuck et al., 2016*; *Schuck et al., 2018*). Our findings align with these reports, highlighting the OFC's role in maintaining internal representations of auditory sequences and generating robust PE signals when deviations occur.

## Limitations and future directions

There are special challenges in interpreting ERP findings in brain lesion populations (e.g., *Kutas et al., 2012*; *Jaeger and Parente, 2008*). Structural brain pathology linked to post-lesion changes in neural tissue and anatomy can introduce variations in electrical activity conduction and alter current flow patterns (*Jaeger and Parente, 2008*; *Løvstad and Cawley, 2011*; *Voytek et al., 2010*). To conclude that ERP differences between patient and control groups reflect functional disturbance in particular cognitive processes, and not primarily effects of structural brain damage, it is useful to demonstrate that they are specific to certain ERP components/stages of information processing and task conditions (*Kutas et al., 2012*; *Swaab, 1998*). The altered ERP responses in the present study were limited to specific task conditions and did not manifest uniformly across all data. This condition-dependent pattern suggests that the observed group differences are related to the specific cognitive processes engaged during those task conditions, rather than being a global artifact of volume conduction. Additionally, the latency differences in scalp potentials observed particularly during the processing of local + global prediction violation further support the notion that these variations reflect genuine differences in cognitive processing (*Hämäläinen et al., 1993*).

Another constraint of our study is the heterogeneity of the LPFC lesion control group, characterized by diverse lesion locations and sizes along the anterior-posterior axis of the LPFC, which could obscure specific functional correlations when compared to the relatively more homogeneous OFC lesion group. Nonetheless, the LPFC group is valuable as it acts as a broader control for assessing the general effects of frontal brain damage, and by contrasting the LPFC group's effects with those of the OFC group, we gain insight into the anatomical specificity of the cognitive processes affected. An additional challenge with focal lesion studies is to establish large patient cohorts. The group size of our study, which is relatively large compared to other studies of focal PFC lesions, does not allow us to perform exploratory lesion-symptom mapping analyses. A larger patient sample is required to

**Table 2.** Characteristics of lesions to the orbitofrontal cortex (OFC).

| | Etiology | Lesion size (cm³) | | | BA (left hemisphere) | BA (right hemisphere) |
|---|---|---|---|---|---|---|
| OFC | | Total | L | R | | |
| 1 | Olfactory meningioma | 43.0 | 23.2 | 19.8 | 10, 11 | 10, 11 |
| 2 | Traumatic brain injury | 24.9 | 6.4 | 18.5 | 11 | 10, 11, 47 |
| 3 | Traumatic brain injury | 157.4 | 59.8 | 97.6 | 8–11, 32, 45–48 | 6, 8–11, 24, 32, 44–48 |
| 4 | Olfactory meningioma | 117.9 | 56.4 | 61.5 | 9–11, 32, 46, 47 | 10, 11, 32, 45–47 |
| 5 | Olfactory meningioma | 6.6 | 3.2 | 3.4 | 11 | 11 |
| 6 | Olfactory meningioma | 8.6 | 3.1 | 5.4 | 11 | 10, 11 |
| 7 | Olfactory meningioma | 8.8 | 1.3 | 7.5 | 11 | 11, 47 |
| 8 | Olfactory meningioma | 3.7 | 3.7 | 0 | 10, 11 | _ |
| 9 | Olfactory meningioma | 85.7 | 55.1 | 30.6 | 9–11, 25, 32, 46, 47 | 10, 11, 47 |
| 10 | Olfactory meningioma | 109.0 | 48.8 | 60.3 | 10, 11, 32, 46, 47 | 9–11, 32, 45–47 |
| 11 | Low-grade glioma | 6.4 | 0 | 6.4 | _ | 10, 11 |
| 12 | Olfactory meningioma | 32.6 | 10.1 | 22.5 | 11 | 10, 11, 25 |

Etiology, size (L, left; and R, right hemisphere), and affected Brodmann areas (BA) for each hemisphere. The sign '_' is used when no lesion was present in a given hemisphere. Lesions that comprise <0.2 cm3 in any given BA are not reported.

draw conclusions about specific OFC subregions' critical roles in PE detection and allow statistical approaches to lesion sub-classification and brain-behavior analysis (e.g., voxel-based lesion-symptom mapping; *Bates et al., 2003*; *Lorca-Puls et al., 2018*). Moreover, the moderate sample size in this study may result in inadequate statistical power to detect effects of OFC lesions in the behavioral performance (i.e., counting of global deviants) despite the altered neurophysiological responses. An effect of lesions on behavioral performance would have strengthened the claim of altered high-level predictive processing. Future studies exploring behavioral nuances of the paradigm, for example, measuring reaction times for correct deviant detection, might uncover lesion effects in participants' deviant detection performance.

# Materials and methods

## Participants

In total, 12 patients with lesions in the OFC and 14 healthy control participants (CTR) were enrolled in the experiment. We also included a lesion control group, which consisted of 10 patients with unilateral lesions to the LPFC; four in the left and six in the right hemisphere (see *Figure 1—figure supplements 2 and 3* for LPFC lesion reconstructions). Among the OFC group, 10 had bilateral damage and 2 had unilateral damage. All patients were in the chronic phase of recovery, that is, at least 2 years' post-tumor resection or trauma. Details about the OFC lesions are provided in *Table 2* and *Figure 1—figure supplement 1*, while details about the LPFC lesions are available in *Figure 1—figure supplement 4* and *Supplementary file 1e*. CTRs were recruited by advertisement and personal contact, whereas patients were recruited through the Department of Neurosurgery at Oslo University Hospital. Inclusion of patients was based on the presence of focal frontal lobe lesions as indicated on pre-existing structural CT and/or MRI scans.

The two study groups did not differ significantly regarding sex, age, or years of education. IQ was estimated based on the Verbal Comprehension and Matrices subtests of the Wechsler Abbreviated Scale of Intelligence (WASI) (*Wechsler, 1999*). The Digit Span test from the Wechsler Adult Intelligence Scale Third Edition (WAIS-III) (*Wechsler and Psychological Corporation, 1997*) was included as a measure of auditory memory span and WM. Verbal learning and memory were assessed with the California Verbal Learning Test Second Edition (CVLT-II) (*Delis et al., 2000*). Two tests from the Delis-Kaplan Executive Function System (D-KEFS) (*Delis, 2001*) were included: the Trail Making Test

**Table 3.** Demographics and neuropsychological performance measures per group.

| Demographics | CTR | SD | OFC | SD | *F*-value | p-value | Stat. |
|---|---|---|---|---|---|---|---|
| N | 14 | | 12 | | | | |
| Gender (females: males) | 8:6 | | 8:4 | | | | |
| Age years (range) | 47.6 (34–66) | 10.3 | 47.9 (27–61) | 11.7 | 0.002 | 0.96 | ns |
| Education years (range) | 16.1 (13-21) | 2.0 | 15 (9–21) | 3.1 | 1.30 | 0.27 | ns |
| **Neuropsychological tests** | | | | | | | |
| Total IQ | 115.4 | 10.3 | 112.2 | 8.5 | 0.73 | 0.40 | ns |
| Digit Span total | 14.8 | 2.9 | 15 | 3.8 | 0.013 | 0.91 | ns |
| Digit Span – forward | 8.5 | 1.5 | 8.8 | 2.1 | 0.09 | 0.77 | ns |
| Digit Span – backward | 6.3 | 1.8 | 6.3 | 2.2 | 0.005 | 0.94 | ns |
| Trail Making Test (TMT) | | | | | *U*-value | | |
| TMT 2 – number sequencing | 30.6 | 10.1 | 33.8 | 14.6 | 92.00 | 0.71 | ns |
| TMT 3 – letter sequencing | 27.9 | 10.5 | 30.3 | 9.9 | 97.50 | 0.49 | ns |
| TMT 4 – number-letter switching | 73.8 | 27.1 | 72.9 | 35.4 | 72.00 | 0.56 | ns |
| Color-Word Interference Test (CWIT) | | | | | | | |
| CWIT 1 – color naming | 31.2 | 6.1 | 30.3 | 4.4 | 75.00 | 0.89 | ns |
| CWIT 2 – word reading | 22.4 | 3.3 | 22.1 | 3.7 | 69.00 | 0.65 | ns |
| CWIT 3 – inhibition | 52.0 | 9.0 | 52.3 | 11.0 | 76.50 | 0.94 | ns |
| CWIT 4 – inhibition/ switching | 58.2 | 11.9 | 60.8 | 18.3 | 74.50 | 0.85 | ns |
| California Verbal Learning Test (CVLT-II) | | | | | | | |
| Total learning trial 1–5 | 57.7 | 12.4 | 51.6 | 8.5 | 49.50 | 0.12 | ns |
| Short-term free recall | 15.2 | 1.2 | 14.8 | 1.4 | 66.50 | 0.54 | ns |
| Long-term free recall | 13.7 | 2.6 | 13.1 | 2.2 | 66.00 | 0.54 | ns |

Comparison of the age, years of education, IQ, and Digit Span Test between the two groups (one-way ANOVA). Comparison of the non-normally distributed raw test scores, Trail Making Test (TMT), Color-Word Interference Test (CWIT) and the California Verbal Learning Test 2nd Edition (CVLT-II) between the two groups (non-parametric independent samples Mann–Whitney *U*-test). Values given are means, with standard deviation (SD). CTR, healthy control group; OFC, group with lesion to the orbitofrontal cortex; ns, the statistical test was not significant.

(TMT), which involves visual scanning, processing speed, and WM, and the Color-Word Interference Test (CWIT), which measures processing speed, inhibition of cognitive interference (i.e., the classical Stroop effect), and mental switching. Group means and statistical comparisons on neuropsychological test measures are reported in *Table 3*. The OFC group did not differ significantly from controls on any of the neuropsychological measures. The lesion control group (i.e., LPFC) also did not differ significantly from the control group regarding sex, age, years of education, or any of the neuropsychological measures (*Supplementary file 1f*).

All participants gave written informed consent before participating in the study. Healthy controls received 400 NOK (approximately 50 USD) for participation in the entire research project (neuropsychological assessment, EEG recording, and MRI scanning). Patients participated in conjunction with clinical follow-ups at the hospital's outpatient clinic. Their travel and accommodation expenses were covered. The study design and protocol were approved by the Regional Committees for Medical and

Health Research Ethics, South-East Norway, as part of a larger study. The study was conducted in accordance with the principles stated in the Declaration of Helsinki.

## Lesion mapping

Lesion mappings were based on structural MRI scans obtained after study inclusion and verified by the neurologist and the neurosurgeon in the research group (RTK and TRM). Lesions were manually outlined on fluid-attenuated inversion recovery (FLAIR) sequence images (1 × 1 × 1 mm³ resolution) for each participant's brain using MRIcron2 (https://www.mccauslandcenter.sc.edu/mricro/mricron/). High-resolution T1-weighted images were used to help determine the borders of the lesions when required. Each participant's brain was extracted from the T1 image using the FSL Bet algorithm (FSL3) and then normalized to the Montreal Neurological Institute MNI-152 template space using the Statistical Parametric Mapping software (SPM12: https://www.fil.ion.ucl.ac.uk/spm/) unified segmentation and normalization procedures, while including the drawn lesions as masks. In addition, the transformation matrix was applied to the individual participant's FLAIR and lesion mask images. *Figure 1* depicts the aggregate lesion reconstructions for the OFC group and the average percentage of damaged tissue within each Brodmann area (BA) per hemisphere (see *Figure 1—figure supplement 1* for individual lesion reconstructions).

## The local–global paradigm and procedures

Two tones composed of three sinusoidal tones (tone A: 440, 880, and 1760 Hz; tone B: 622, 1244, and 2488 Hz) were synthesized. Each tone was 50 ms long, with 7 ms rise and fall times. Sequences of four or five such tones were delivered with a fixed stimulus onset asynchrony (SOA) of 150 ms. Each sequence's SOA was randomly drawn from a uniform distribution between 1350 and 1650 ms. Three different types of sequences were presented: (i) sequences comprised by five identical tones AAAAA or BBBBB (jointly denoted by **xx**), (ii) sequences comprised by four identical tones and a fifth different tone AAAAB or BBBBA (jointly denoted by **xy**), or (iii) sequences comprised by four identical tones AAAA_ or BBBB_ (jointly denoted by **xo**). Two block types were defined in the present set of analyses: regular (xX) and irregular (xY). Each block of trials started with the repetition of 20 identical sequences of tones to establish the block's global rule, followed by 100 test trials. **Block xX**: 75% **xx** sequences referred to as **xx│xX** trials and 12,5% **xy** sequences referred to as **xy│xX** trials. **Block xY**: 75% **xy** sequences referred to as **xy│xY** trials and 12,5% **xx** sequences referred to as **xx│xY** trials. 12.5% of **xo** sequences were included in Block xX and Xy (see *Figure 2B*).

The experiment included two experimental sessions with 12 blocks in total (six blocks of trials in each session, and each block type was presented twice where tones A and B were swapped, making a total of 1440 trials). The experimental blocks (of ~3 min duration each) were randomized across sessions and participants, except that the first block of each session was always Block xX. The paradigm enables the statistical contrast of trials that have the same physical stimulus properties but differ in their stimulus transition probabilities and therefore in their predictability. The task does not entail any symbolic reward and no performance feedback is provided. Sound presentation was controlled with MATLAB (R2018a, MathWorks Inc, Natick, MA), using the Psychophysics Toolbox version 3 (*Kleiner et al., 2007*).

Participants were seated comfortably in a Faraday-shielded room in front of an LCD monitor with a 60 Hz refresh rate placed at a distance of ~70 cm from the participant while presented with auditory stimuli. Stimuli were delivered through speakers on the side of the screen at a comfortable volume. Participants were instructed to attend to the auditory stimuli and count any rare/uncommon sequences. At the end of each block, participants reported this count in a data sheet before continuing the experiment. Hence, we expected participants to attend to and extract the global rule that characterized deviant sequences.

## EEG acquisition and preprocessing

EEG was recorded at a 1024 Hz sampling rate using a 64-channel Active Two system (BioSemi, Amsterdam, Netherlands) with active electrodes placed in accordance with the International 10–20 system (*Chatrian et al., 1985*). In addition, six external electrodes were used, including two electrodes placed above and below the right eye and two placed at the right and left outer canthus

(vertical and horizontal EOG channels, respectively). The last two electrodes were placed on the right and left earlobes for offline re-referencing.

We used the FieldTrip toolbox (*Oostenveld et al., 2011*) for MATLAB (R2018a, MathWorks Inc) for offline EEG data processing. EEG data were re-referenced to averaged earlobes, and the linear trend was subtracted. The continuous EEG data were then high-pass filtered back and forward (zero-phase) with an infinite impulse-response (IIR) Butterworth filter (order: 3), half-amplitude cutoff at 0.01 Hz. Spectral interference by power line noise was ameliorated by using the method of spectrum interpolation (*Leske and Dalal, 2019*), targeting the line noise frequency (50 Hz) and its first four harmonics. The continuous data were visually inspected, and noise-contaminated channels and segments were identified (e.g., large muscle artifacts). The sample information of the noisy segments was saved for later rejection of epochs overlapping with these segments. Bad channels were removed before running an independent component analysis (ICA). The ICA was used to identify and then manually remove blinks and horizontal eye movements (ocular components) in the non-epoched data. Rejected channels were subsequently interpolated from the neighboring electrodes using spherical spline interpolation (*Perrin et al., 1989*).

To ensure the validity of the neural data analysis, potential sources of bias were assessed between the healthy control participants and the OFC lesion group. Specifically, no significant differences were observed between the two groups in terms of the number of noisy channels, the number of clean trials (i.e., trials remaining after removing the noisy segments from the data), or the number of blinks across the task blocks and the experimental conditions (see *Supplementary file 1g*).

## Data analysis

### Behavioral analysis

Behavioral data were collected in the form of a written report of the number of rare/uncommon sequences of tones detected (i.e., global deviants) per block. For each participant, we computed and reported the average percentage of errors over blocks, which is the deviation (positive or negative) relative to the actual number of presented global deviants.

### Statistical analysis of ERPs

The preprocessed data were segmented into epochs of –2400 ms to 1800 ms relative to the onset of the last tone presentation and used for further analyses. Epochs in the habituation phase (i.e., first 20 trials) of all blocks were excluded from further analysis. ERPs were calculated by averaging the stimulus-locked epochs for each condition. Prior to averaging, epochs were downsampled to 512 Hz and baseline corrected relative to the mean activity –600 to –150 ms before the onset of the last tone.

To investigate deviance processing, we hypothesized an internal model with two hierarchical levels forming predictions and generating PEs that interact within and across levels – based on the hierarchical predictive coding model of local and global novelty proposed by *Chao et al., 2018*. In accordance with the predictive coding theory, on control (xx|xX) trials, the fifth tone 'x' is predicted by a low-level prediction, and thus no PE should be elicited. In contrast, on local deviant (xy|xY) trials, enhanced low-level PEs arise because the last tone 'y' violates the transition probability established by the prior sequence of xxxx. High-level predictions anticipate these local violations, and thus high-level PEs are not expected to be elicited. On global deviant (xx|xY) trials, presumably only high-level PEs occur, caused by the unpredicted absence of tone 'y', or more precisely, the absence of low-level PEs is unpredicted by the high level. Finally, on local + global deviant (xy|xX) trials, PEs arise at a low level of processing since the expected tone 'x' is replaced by the tone 'y'. In addition, these PEs, not anticipated by the global rule, activate a higher hierarchical level and elicit high-level PEs.

EEG-derived ERPs for the four conditions of interest (i.e., control, local deviant, global deviant, and local + global deviant) were separately averaged to assess the neuronal markers of deviance processing. Furthermore, we evaluated three main condition contrasts. By contrasting control (xx|xX) and local deviant (xy|xY) trials, we isolated the **local deviance** or low-level PE response, which arises when only local regularities are violated (i.e., a local deviance response predicted by the global rule). Contrasting control and global deviant (xx|xY) trials isolated the **global deviance**, or high-level PE response, unaffected by local deviancy (i.e., a response to a stimulus that is unpredicted by the global rule but predicted at the local level). Finally, by contrasting control and local + global deviant (xy|xX) trials, we examined the **local + global deviance** response, or the combined effect of low- and

high-level PE, that occurs when both local and global regularities are violated (i.e., a local deviance response that is unpredicted by the global rule) (*Figure 2C*).

Pairs of experimental conditions were compared using a non-parametric cluster-based permutation method implemented in the FieldTrip toolbox, addressing the multiple comparisons issue (*Maris and Oostenveld, 2007*). Cluster statistics were obtained by summing the *t*-values that were adjacent in space and time above and below a threshold level (two-sided *t*-test with alpha level 0.05). The cluster p-value was then obtained by comparing the cluster statistic to a null distribution statistic obtained by randomly switching condition labels within average participant responses 1000 times. The clusters were considered as significant when the sum of *t*-values exceeded 97.5% or were below 2.5% of the null distribution. First, cluster-based permutation dependent samples *t*-tests were computed between 0 and 800 ms after the onset of the fifth tone for all the main condition comparisons. Tests were computed for the CTR and OFC groups separately. In a second step, to check for differences in the ERPs between the two main study groups, we ran the same cluster-based permutation approach contrasting each of the four conditions of interest between the groups using independent samples *t*-tests. The cluster-based permutation independent samples *t*-tests were computed in the latency range of each component, which was determined based on the maximum range for both groups combined. The latency range for each group and component was based on the time frames derived from the statistical analysis of task condition contrasts.

We estimated component latencies using the 50% area latency method (*Luck, 2014*; *Liesefeld, 2018*), which calculates the time point when an ERP component reaches 50% of its area under the curve. Area latency reflects the median latency, which is considered a more reliable measure than the traditional peak latency (*Luck, 2014*; *Liesefeld, 2018*). The component area was defined as the entire time window of the ERP component. To avoid missing component activity in individuals, we selected a broad time window (i.e., local deviance MMN: 50–150 ms and P3a: 140–350 ms; local + global deviance MMN: 50–220 ms for the CTR and 50–250 ms for the OFC group, P3a: 180–450 ms for the CTR and 180–500 ms for the OFC group, P3b: 380–800 ms; global deviance P3b: 380–800 ms). To assess group differences in component latencies, independent samples *t*-tests were performed for each component of interest and each condition for the five midline electrodes (i.e., Fz, FCz, Cz, CPz, and Pz). To account for the multiple comparisons problem, we performed FDR correction (*Benjamini and Hochberg, 1995*) across channels.

To provide clarity regarding the nature of the observed ERP group differences (i.e., whether they are amplitude differences or outcomes of latency variations), we conducted complementary analyses on mean amplitudes of the ERP components for the conditions where significant group differences were observed. The mean amplitudes were calculated centered around the individual peaks for each component (see *Supplementary file 1d and e* ).

## Conclusion

We tested the role of the OFC in detecting violations of prediction (i.e., PEs) at two hierarchical levels of task structural complexity. Our critical finding is that low-level PEs (i.e., processing of stimuli that are unpredicted at the local level) and combined low- and high-level PEs (i.e., processing of stimuli that are unpredicted at both the local and global levels) were impacted by the OFC lesion as reflected in the altered MMN and P3a components. We suggest that the OFC likely contributes to a top-down predictive process that modulates the deviance detection system in lower sensory areas. The study sheds new light on the poorly explored involvement of the OFC in hierarchical auditory predictive processing.

## Acknowledgements

We are very thankful to all the patients and healthy controls who participated in this study. We acknowledge Professor Per Kristian Hol, Intervention Center at Oslo University Hospital, for his valuable help with clinical evaluation of the MRI scans of patients and healthy controls.

# Additional information

## Funding

| Funder | Grant reference number | Author |
|---|---|---|
| Research Council of Norway | 240389 | Torstein R Meling<br>Tor Endestad<br>Anne-Kristin Solbakk |
| Research Council of Norway | 314925 | Alejandro Omar Blenkmann |
| Research Council of Norway | RITMO 262762 | Tor Endestad<br>Anne-Kristin Solbakk |
| Research Council of Norway | RITPART 274996 | Tor Endestad<br>Anne-Kristin Solbakk |
| National Institute of Neurological Disorders and Stroke | NINDS R37NS21135 | Robert T Knight |
| National Institute of Neurological Disorders and Stroke | Conte Center PO 518 MH109429 | Robert T Knight |

The funders had no role in study design, data collection and interpretation, or the decision to submit the work for publication.

## Author contributions

Olgerta Asko, Conceptualization, Data curation, Software, Formal analysis, Validation, Investigation, Visualization, Methodology, Writing – original draft, Project administration, Writing – review and editing; Alejandro Omar Blenkmann, Conceptualization, Data curation, Software, Formal analysis, Supervision, Funding acquisition, Validation, Investigation, Methodology, Writing – original draft, Project administration, Writing – review and editing; Sabine Liliana Leske, Software, Formal analysis, Validation, Investigation, Methodology, Writing – review and editing; Maja Dyhre Foldal, Investigation, Writing – review and editing; Anais LLorens, Conceptualization, Software, Investigation, Writing – review and editing; Ingrid Funderud, Conceptualization, Data curation, Investigation, Methodology, Project administration, Writing – review and editing; Torstein R Meling, Tor Endestad, Conceptualization, Resources, Data curation, Supervision, Funding acquisition, Investigation, Methodology, Writing – review and editing; Robert T Knight, Conceptualization, Supervision, Investigation, Methodology, Writing – review and editing; Anne-Kristin Solbakk, Conceptualization, Resources, Data curation, Formal analysis, Supervision, Funding acquisition, Validation, Investigation, Methodology, Writing – original draft, Project administration, Writing – review and editing

## Author ORCIDs

Olgerta Asko ⓘ https://orcid.org/0000-0003-3608-3621
Alejandro Omar Blenkmann ⓘ http://orcid.org/0000-0002-8671-2214
Anais LLorens ⓘ http://orcid.org/0000-0002-5842-6798

## Ethics

The study involving human participants was reviewed and approved by Regional Committees for Medical and Health Research Ethics, South-East Norway (REK. number: 2014/381) as part of a larger study. The study was conducted in accordance with the principles stated in the Declaration of Helsinki. All participants provided written informed consent and received compensation for their participation.

## Decision letter and Author response

Decision letter https://doi.org/10.7554/eLife.86386.sa1
Author response https://doi.org/10.7554/eLife.86386.sa2

## Additional files

### Supplementary files

• MDAR checklist

• Supplementary file 1. Additional measurements. Table a: mean amplitude centered ±25 ms around the individual peaks for the MMN and P3a components elicited for the local deviance response for the two groups (CTR vs. OFC). Table b: mean amplitude centered ±40 ms around the individual peaks for the MMN and P3a components elicited for the local + global deviance response for the two groups. Table c: 50% area latency for the MMN and P3a components elicited for local and local + global deviance response for the healthy control participants. Table d: 50% area latency for the MMN and P3a components for the difference wave (local + global minus local deviance response) for the two groups. Table e: characteristics of lesions to the LPFC lesion group (LPFC). Table f: demographics and neuropsychological performance measures per group (CTR vs. LPFC). Table g: Additional measurements that could bias the neural data for the CTR and OFC group (e.g., number of blinks, noisy channels, and noisy trials).

### Data availability

We do not have permission from the Regional Committees for Medical and Health Research Ethics (REC) to share clinical data publicly. The conditions of the ethical approval of this study do not permit public archiving of de-identified data. Neither the patients nor the healthy control participants have consented to making their data publicly available. The reasons for the restrictions concerning public sharing of clinical data is that the patient samples are small given the relative rareness of individuals with focal brain lesions, and thus the constellation of demographic and clinical information results in an increased risk of patients being identified. Interested researchers seeking access to the original de-identified data supporting the claims in this paper would have to submit a short study plan of the proposed research to the PI of the project and lesion registry, Anne-Kristin Solbakk (a.k.solbakk@ psykologi.uio.no). The study plan would be evaluated by the project PI, the head of the Department of Neurosurgery at Oslo University Hospital, and the head of research at the Department of Psychology. Next, the PI would ask REC for permission to share de-identified data with the researcher/institution. After REC approval, the head of research at the Department of Psychology, the Data Protection Officer at Oslo University Hospital, and the other interested party would sign data transfer agreements, before data transfer would take place. The data can be accessed and used only for academic purposes. Commercial research cannot be performed on the data. Materials for the Experimental scripts and task stimuli, custom analysis code, and a processed version of the dataset are available at https://osf.io/f9m76/.

The following dataset was generated:

| Author(s) | Year | Dataset title | Dataset URL | Database and Identifier |
|---|---|---|---|---|
| Asko O, Blenkmann AO, Leske SL, Foldal MD, Lorens AL, Funderund I, Meling TR, Endestad T, Solbakk AK | 2023 | Altered hierarchical auditory predictive processing after lesions to the orbitofrontal cortex | https://doi.org/10.17605/OSF.IO/F9M76 | Open Science Framework, 10.17605/OSF.IO/F9M76 |

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
