## [Editor Report]

This important study demonstrates that the orbitofrontal cortex is involved in the detection of local auditory prediction errors. The methods and procedures are convincing, although the precise functional meaning of the reported effects remains to be specified. This work will be of interest to neuropsychologists and cognitive neuroscientists working on the prefrontal cortex, predictive processing, auditory perception, and electrophysiology.

---

## [Decision Letter]

**Decision letter after peer review:**

Thank you for submitting your article "Altered hierarchical auditory predictive processing after lesions to the orbitofrontal cortex" for consideration by *eLife*. Your article has been reviewed by 3 peer reviewers, and the evaluation has been overseen by Maria Chait as Reviewing Editor and Andrew King as the Senior Editor.

We all agreed this is an interesting, well performed study. However, there are some issues with analysis and interpretation that we hope can be addressed in a revision:

Analysis:

1) Justification for choice of the time frames to test for group differences of ERP components must be clarified. Additionally, complementary analyses on mean or peak amplitudes could resolve the issue re whether it is just the latency of the relevant components that is affected.

2) Confirm that the lesion and control groups do not differ on measures that could inadvertently bias the neural data. This includes various issues with noise and behavioural performance/task difficulty/attention.

Interpretation:

1) A difference between the lateral PFC group and controls is expected based on previous literature and implication of this brain area in predictive coding. That no differences are seen here needs to be justified more carefully.

2) The main conclusion re role of OFC is not clear given that there is no difference between patient groups and controls on the global task therefore hierarchical predictive processing itself seems to be unaltered, even though its neural correlates were different. The functional meaning of the EEG findings must be discussed more clearly and carefully.

3) In general the introduction and discussion could be shortened to focus more clearly on the key results/interpretation.

*Reviewer #1 (Recommendations for the authors):*

I very much enjoyed this manuscript. I therefore only have three main areas that I would encourage the authors to reflect on.

As noted above, and for which I will expand on here, it was not clear to me how the time frames to test for group differences were defined. On page 14 it says "The cluster-based permutation independent samples t-tests were computed in the latency range of each component, which was based on the statistical analysis of the CTR and OFC group condition contrasts." Do these refer to the latency ranges defined in the next paragraph and if so was it the maximum range for both groups? I would appreciate it if this was clarified. It would be important to reassure the readers that no biases were introduced by sampling latencies dominated by one group compared to another. On this point I noted on page 9 of the supplementary material when referring to the lateral PFC analyses it says "We also tested for group difference in the time range (i.e., 140 to 350 ms, based on the statistical analysis of the CTR group task condition contrasts) of the P3a". What is the justification for this analysis approach? I would be concerned that it introduced biases. When defining time-windows of interest I believe it is common to define components based on the grand averages across all participants. Could the authors clarify precisely methods for defining the time window of interest analysis.

Following from my note in the public review I would appreciate it if the authors were in a position to speculate on how the detection of prediction errors and the assignment of credit to stimuli might (or might not) functionally converge within the OFC.

It is hugely impressive to have recruited such a large group of relatively homogenous lesion patients for this study. Further to my comments in the public review, are there any clues that we can take away to increase the anatomical specificity of the study. For example, is there enough coverage and power to perform any exploratory lesion symptom mapping analyses?

*Reviewer #3 (Recommendations for the authors):*

The labeling of the Local+Global deviant ("xy|xY") seems to be wrong in paragraph 2 of the section "The Local-Global paradigm and procedures" and in figure 2.

---

## [Author Response]

Analysis:1) Justification for choice of the time frames to test for group differences of ERP components must be clarified. Additionally, complementary analyses on mean or peak amplitudes could resolve the issue re whether it is just the latency of the relevant components that is affected.

We greatly appreciate the insightful feedback, which prompted us to refine our analysis approach to better address the temporal aspects of ERP components and enhance the robustness of our findings. Regarding the choice of time frames for testing group differences in ERP components, we recognize the need for a more thorough justification. In the revised version, we included an expanded section that details the rationale behind our time frame selection. See Response to Reviewer #1 Comment 1. Additionally, we acknowledge the value of conducting complementary analyses on mean or peak amplitudes to investigate further whether the observed latency differences are accompanied by corresponding changes in amplitude. Consequently, we included supplementary analyses that explored mean amplitudes for the identified ERP components.

To provide a clearer understanding of the nature of the observed ERP amplitude differences, we conducted complementary analyses on mean amplitudes of the MMN and P3a components on the midline sensors for the conditions where significant group differences were observed (i.e., the Local and the Local + Global Deviance conditions).

For the MMN component elicited by the Local Deviance, we found group amplitude differences on the electrodes AFz (p = 0.021), Fz (p = 0.008), CPz (p = 0.015), and Pz (p < 0.001). Surprisingly, we also found amplitude differences for the P3a component elicited by the Local Deviance on the electrodes AFz (p < 0.001), Fz (p < 0.001), FCz (p < 0.001), and Cz (p = 0.002) that were not observed previously with the cluster-based permutation analysis. For the MMN component elicited by the Local + Global Deviance, our analysis showed group amplitude differences on the electrodes AFz (p = 0.007), FCz (p = 0.051), Cz (p = 0.004), CPz (p = 0.002), and Pz (p < 0.001). However, as the reviewer rightly pointed out, the group differences for the P3a elicited by the Local + Global Deviance seem to be a byproduct of latency differences, as we did not find amplitude differences on any of the midline electrodes. Overall, this complementary analysis shows that the OFC patients had an attenuated MMN/P3a to local level prediction violation, and an attenuated and delayed MMN followed by a delayed P3a to the combined local and global level prediction violation.

These results are presented in the Supplementary File 1, Supplementary Files 1a and 1b and in the section “Results – EEG results” of the main manuscript [page 12, paragraph 2], which states: “To better understand the nature of the ERP group differences revealed by the cluster-based permutation tests, complementary analysis on the mean amplitudes of the MMN and P3a components was conducted. The MMN was defined as the most negative peak in a post-stimulus window of 50-150 ms, and the P3a as the most positive peak in a post-stimulus window of 130-310 ms. The mean amplitude was calculated centered ± 25 ms around individual peaks. The independent samples t-tests comparing the distinct components mean amplitudes between the two groups for the midline sensors revealed amplitude differences for the MMN [AFz (P = 0.021), Fz (P = 0.008), CPz (P = 0.015), and Pz (P < 0.001)] and for the P3a [AFz (P < 0.001), Fz (P < 0.001), FCz (P < 0.001), and Cz (P = 0.002)] (Supplementary File 1a).” and in the section “Results – EEG results” of the main manuscript [page 20, paragraph 1], which states: “To provide clarity regarding whether the MMN and P3a group differences revealed by the cluster-based permutation tests are amplitude differences or outcomes of latency variations, complementary analysis on the mean amplitudes of the MMN and P3a components was conducted. The MMN was defined as the most negative peak in a post-stimulus window of 50-250 ms and the P3a as the most positive peak in a post-stimulus window of 150 -350 ms. The mean amplitude was calculated centered ± 40 ms around individual peaks. The independent samples t-tests comparing the distinct components mean amplitudes between the two groups for the midline sensors revealed amplitude differences for the MMN [AFz (P = 0.007), FCz (P = 0.051), Cz (P = 0.004), CPz (P = 0.002), and Pz (P < 0.001)], but not for the P3a (Supplementary File 1b). Thus, the group differences for the P3a elicited by the Local + Global Deviance seem to be a byproduct of latency differences.”

We have also added text in the section “Materials and methods – Statistical analysis of event-related potentials” of the main manuscript [page 40, paragraph 3], which states: “To provide clarity regarding the nature of the observed ERP group differences (i.e., whether they are amplitude differences or outcomes of latency variations), we conducted complementary analyses on mean amplitudes of the ERP components for the conditions where significant group differences were observed. The mean amplitudes were calculated centered around the individual peaks for each component (see Supplementary File 1a and 1b)”.

2) Confirm that the lesion and control groups do not differ on measures that could inadvertently bias the neural data. This includes various issues with noise and behavioural performance/task difficulty/attention.

We found this suggestion excellent, and therefore completed a number of tests to ensure that the orbitofrontal cortex (OFC) lesion group and the control group did not differ on measures that could bias the neural data.

We thank the reviewer for this suggestion. We have completed a number of measurements and tests to ensure that the OFC lesion group and the control group did not differ on measures that could affect the neural data. First, we computed the number of bad/noisy channels for each subject and group, and found that the two groups did not differ significantly. Second, we computed the number of trials remaining after removing the noisy segments across conditions for each subject and group, and found no significant differences between the groups. Third, the number of blinks/saccades across conditions for each subject and group showed no significant group differences. Altogether, the results indicate that the neural differences observed in our study arose because of the specific lesion effect.

These additional EEG measures and the statistical test results are included in the Supplementary File 1, Supplementary File 1g. We have also added text in the section “Materials and methods – EEG acquisition and pre-processing” of the main manuscript [page 38, paragraph 2], which states: “To ensure the validity of the neural data analysis, potential sources of bias were assessed between the healthy control participants and the OFC lesion patients. Specifically, no significant differences were observed between the two groups in terms of the number of noisy channels, the number of clean trials (i.e., trials remaining after removing the noisy segments from the data), or the number of blinks across the task blocks and the experimental conditions (see Supplementary File 1g).”

Thank you for pointing this out. Indeed, the experimental blocks differ in the number of deviant tones and therefore in the task difficulty. Thus, it is a very good suggestion to look for behavioral performance differences across the different blocks. In the present set of analyses, two block types were used: Regular (xX) and Irregular (xY). In regular blocks, where the repeated sequence is xxxxx, participants were required to count the rare/uncommon sequences, i.e., xxxxy and xxxxo. In irregular blocks, where the repeated sequence is xxxxy, participants were required to count the rare/uncommon sequences, i.e., xxxxx and xxxxo. We have now updated the behavioral analysis. First, by excluding the omission block’s counting performance, and second, by calculating the counting performance separately for the two blocks. The new behavioral analysis revealed that participants from both groups performed better in the irregular block compared to the regular block. However, there was no statistically significant difference between the counting performances of the two groups.

The new results are reported on page 11 of the main manuscript, section “Results – Behavioral performance”: “Participants performed the task properly with an average error rate of 9.54% (SD 8.97) for the healthy control participants and 10.55% (SD 6.18) for the OFC lesion patients. There was no statistically significant difference between the counts of rare tone sequences of the two groups [F(24) = 0.11, P = 0.75]. Participants from both groups had a trend-level better performance in the irregular block (CTR: 8.39 ± 8.24%; OFC: 7.50 ± 7.34%) compared to the regular block (CTR: 10.69 ± 11.36%; OFC: 13.60 ± 10.97%) [F(24) = 3.55, P = 0.07]. There was no block X group interaction effect [F(24) = 0.73, P = 0.40].”

Interpretation:1) A difference between the lateral PFC group and controls is expected based on previous literature and implication of this brain area in predictive coding. That no differences are seen here needs to be justified more carefully.

We thank the reviewers and editors for pointing this out. We also expected differences between the lateral prefrontal cortex (PFC) group and controls considering previous literature implicating this brain area in predictive coding. We understand the importance of addressing this point more carefully, and therefore, provide a more detailed explanation for the lack of observed differences in our study.

We thank the reviewer for raising this crucial issue. We recognize the importance of addressing the lack of neurophysiological differences between the lateral PFC lesion group and the control group. First, it is important to clarify that the lateral PFC lesion control group was initially included not as a control for specific lateral PFC lesions but rather a broader control group to account for potentially general effects of frontal brain damage. However, considering that previous studies have implicated specific areas of the lateral PFC (e.g., inferior frontal gyrus; IFG) in predictive processing, we also think that a more thorough justification of these null findings is needed.

Intracranial EEG studies examining local and global level prediction error detection pointed to the role of inferior frontal gyrus (IFG) as a frontal source supporting top-down predictions in MMN generation (Dürschmid et al., 2016; Nourski et al., 2018; Phillips et al., 2016; Rosburg et al., 2005). However, other intracranial studies reported unclear (Bekinschtein et al., 2009) or weak (Dürschmid et al., 2016) frontal MMN effects. El Karoui et al. (2015) observed late ERP responses in the lateral PFC related to global deviants but no MMN to local deviants, and it was not clear where in the PFC these responses occurred, not showing responses in the IFG. Additionally, studies employing dynamic causal modeling of MMN consistently modeled frontal sources in the IFG region (Garrido et al., 2008; Garrido et al., 2009; Phillips et al., 2015). A review by Deouell (2007) highlighted the potential contributions of both IFG and middle frontal gyrus to MMN generation, suggesting that the specific source might vary depending on characteristics of the deviant stimuli, such as pitch or duration.

In Alho et al. (1994) lesion study, diminished MMN to local-level deviants was found after lesion to the lateral PFC, with the lesion cohort exhibiting a hemisphere ratio of 7/3 for left and right hemispheres, respectively, which is different from our cohort's ratio of 4/6. Furthermore, all individuals in that study had infarcts in the middle cerebral artery, resulting in a more uniform lesion location compared to our cohort. Notably, the lesions observed in our lateral PFC group appeared to be situated in more superior brain regions and towards the MFG compared to the predominantly reported involvement of the IFG in previous studies. Another factor that might contribute to the lack of significant effects is the heterogeneity of the lesions in our lateral PFC group (see Supplementary Figures 2, 3 and 4). Especially for the left hemisphere cohort, the individual lesions did not share a consistent anatomical location. The right hemisphere cohort had a greater lesion overlap, but overall, the lesions were not centered in the IFG area with highest overlap being in the MFG area. This distinction in lesion location might contribute to the absence of effects observed in our study.

Regarding the global effect, often reflected in the P300 component, it appears that the neural sources responsible for processing global deviance exhibit a more distributed pattern. This means that the brain regions involved in detecting and processing global deviations may not be as localized or concentrated as those implicated in local deviance processing. Given that the neural mechanisms underlying global deviance detection and processing are likely to involve a wider network of brain regions, they may be less susceptible to disruptions caused by focal lesions in the lateral PFC.

In response to your comment, we have expanded the “Discussion” to address this point by adding a new section titled “Lack of findings in the lateral PFC lesion group” [page 29]. In this section, we first present some of the findings implicating specific areas of the lateral PFC in the generation of MMN and in predictive processing, and then offer an account of the potential reasons behind the lack of neurophysiological differences between the lateral PFC and control groups.

We appreciate the reviewer's comment and want to acknowledge that another reviewer raised this concern previously. We have expanded the “Discussion” to address this point by adding a new section titled “Lack of findings in the lateral PFC lesion group” [page 29]. In this section, we first present some of the findings implicating specific areas of the lateral PFC in the generation of MMN and in predictive processing, and then offer an account of the potential reasons behind the lack of neurophysiological differences between the lateral PFC and control groups.

2) The main conclusion re role of OFC is not clear given that there is no difference between patient groups and controls on the global task therefore hierarchical predictive processing itself seems to be unaltered, even though its neural correlates were different. The functional meaning of the EEG findings must be discussed more clearly and carefully.

We understand the need for a more careful discussion of the functional meaning of our EEG findings. While the behavioral results (i.e., detecting global deviants) did not reveal significant differences between the groups, the neural correlates observed for the Local and Local + Global condition, where the influence of the Global rule is present, suggest that there might be subtle differences in how the OFC contributes to hierarchical predictive processing. We have revisited the Discussion section of our manuscript to more precisely articulate the potential functional implications of the observed EEG differences.

3) In general the introduction and discussion could be shortened to focus more clearly on the key results/interpretation.

We appreciate the suggestion for improving the clarity and focus of our manuscript. We made revisions to streamline both the introduction and Discussion sections while still providing essential context and insights into our findings. Our primary goal in the introduction was to establish the background and rationale for the study, and we have restructured it to ensure a more concise presentation of the key motivations and research gaps. Similarly, in the Discussion section, we emphasized the most significant findings and interpretations while eliminating redundant discussions.

Reviewer #1 (Recommendations for the authors):I very much enjoyed this manuscript. I therefore only have three main areas that I would encourage the authors to reflect on.

We thank the reviewer for careful consideration of our work and are delighted to hear that the reviewer enjoyed reading our manuscript. We have considered all the comments and addressed the areas identified for reflection and improvement.

1. As noted above, and for which I will expand on here, it was not clear to me how the time frames to test for group differences were defined. On page 14 it says "The cluster-based permutation independent samples t-tests were computed in the latency range of each component, which was based on the statistical analysis of the CTR and OFC group condition contrasts." Do these refer to the latency ranges defined in the next paragraph and if so was it the maximum range for both groups? I would appreciate it if this was clarified. It would be important to reassure the readers that no biases were introduced by sampling latencies dominated by one group compared to another. On this point I noted on page 9 of the supplementary material when referring to the lateral PFC analyses it says "We also tested for group difference in the time range (i.e., 140 to 350 ms, based on the statistical analysis of the CTR group task condition contrasts) of the P3a". What is the justification for this analysis approach? I would be concerned that it introduced biases. When defining time-windows of interest I believe it is common to define components based on the grand averages across all participants. Could the authors clarify precisely methods for defining the time window of interest analysis.

We thank the reviewer for the helpful comment. We recognize the need for a clearer definition of time frames for testing group differences in the ERP components and apologize for any ambiguity in the previous version of the manuscript.

Regarding the time frames to test for group differences of ERP components for the OFC and control groups, they were determined based on the combined maximum range for both groups. The time range for each group and each ERP component was derived from the statistical analysis of the condition contrasts run for each group. For instance, for the Local Deviance MMN, the condition contrast (i.e., Control condition versus Local Deviance condition) for the CTR group revealed a MMN component from 67 to128 ms, while the same condition contrast for the OFC group revealed a MMN from 73 to131 ms. The time frame used for the group comparison on the MMN time window was 50 to 150 ms to capture component activity for both groups. In the same way, for the Local Deviance P3a, the condition contrast (i.e., Control condition versus Local Deviance condition) for the CTR group revealed a P3a component ranging from 141 to 313 ms, while the same condition contrast for the OFC group revealed a P3a from 145 to 344 ms. The time frame used for the group comparison on the P3a time window encompassed 140 to 350 ms to capture component activity for both groups.

In the “Results – EEG Results” section of the main manuscript, together with the results from the cluster-based permutation independent samples t-tests, we provide the time frames in which the latter were computed for each ERP component. Moreover, in the section “Materials and methods – Data analysis – Statistical analysis of event-related potentials” of the main manuscript [page 40, paragraph 1], we provide a revised description of how the time frames for group differences of ERPs were defined. The revised description states: “In a second step, to check for differences in the ERPs between the two main study groups, we ran the same cluster-based permutation approach contrasting each of the four conditions of interest between the two groups using independent samples t-tests. The cluster-based permutation independent samples t-tests were computed in the latency range of each component, which was determined based on the maximum range for both groups combined. The latency range for each group and component was based on the time frames derived from the statistical analysis of task condition contrasts.”

Regarding the comparisons between the lateral PFC and control groups, they were not based solely on the control group condition contrast. This was miswritten. The approach to define time frames to test for ERP differences between the CTR and the lateral PFC group was the same as the one used to test differences between CTR and OFC groups. We apologize for any confusion this may have caused.

2. I would appreciate it if the authors were in a position to speculate on how the detection of prediction errors and the assignment of credit to stimuli might (or might not) functionally converge within the OFC.

We thank the reviewer for this comment regarding the unexplored conceptual link between our study’s conclusion, which suggests that the OFC facilitates the detection of prediction errors, and the findings of other research that delves into the OFC’s role in assignment of credit to stimuli. We find this comment very interesting and appreciate the opportunity to speculate on the potential functional convergence of these two processes within the OFC.

The OFC is a critical neural hub implicated in learning, decision-making, and adaptive behavior. The detection of prediction errors and the assignment of credit to stimuli are mechanisms linked with the OFC, which play an important role in all these functions (Noonan et al., 2012; Schultz and Dickinson, 2000; Sul et al., 2010; Tobler et al., 2006; Walton et al., 2010; Walton et al., 2011). Prediction errors involve recognizing discrepancies between expected and actual outcomes, which engages the OFC in rapidly updating stimulus valuations to align with newfound information (Holroyd and Coles, 2002; Kakade and Dayan, 2002). Signaling of errors provides a powerful mechanism whereby OFC facilitates adaptive learning and enables the brain to adjust its expectations based on novel experiences (Schultz, 2015; Seymour et al., 2004). Credit assignment, on the other hand, refers to properly identifying the causes of prediction errors. Without proper credit assignment, one might have intact error signaling mechanisms, but lose the ability to learn appropriately. This is especially true when multiple possible antecedents may be related to the error or when past choices have been unpredictable. In such situations, it is important to assign credit to the most recent choice and not get distracted by previous alternatives (Stalnaker et al., 2015).

These mechanisms within the OFC appear interrelated yet distinct. While prediction errors could trigger credit assignment, the OFC's ability to continually assess stimuli's values extends beyond instances of prediction errors. The OFC is involved in continuously evaluating and updating the values of stimuli based on ongoing experiences (Padoa-Schioppa and Assad, 2006; Tremblay and Schultz, 1999). This process enables the brain to learn from both unexpected outcomes and regular, predictable interactions with the environment. In situations where outcomes are not solely determined by prediction errors, the assignment of credit remains important. Complex decision-making involves considering a variety of factors beyond just prediction errors, such as contextual information and long-term consequences. Clarifying the convergence of these mechanisms within the OFC holds profound implications for understanding the intricacies of learning dynamics and the orchestration of adaptive responses to the environment.

While we recognize the value of this discussion, we believe it extends beyond the primary focus of our study. Consequently, we have made the decision not to incorporate it into the current manuscript.

3. It is hugely impressive to have recruited such a large group of relatively homogenous lesion patients for this study. Further to my comments in the public review, are there any clues that we can take away to increase the anatomical specificity of the study. For example, is there enough coverage and power to perform any exploratory lesion symptom mapping analyses?

The reviewer raised an important point here. It would have been interesting to explore this aspect. However, one challenge with focal lesion studies is to establish large patient cohorts. The group size of our study, which is relatively large compared to other studies of focal PFC lesions, does not allow us to perform any exploratory lesion-symptom mapping analyses. A larger patient sample will provide a stronger basis for drawing conclusions about the critical role of a particular OFC subregion to the detection of prediction errors and allow statistical approaches to lesion subclassification and brain-behavior analysis (e.g., voxel-based lesion-symptom mapping (Bates et al., 2003; Lorca-Puls et al., 2018)).

Considering the average percentage of damaged tissue in our study, the medial part of OFC or Brodmann area 11 is affected more by the lesion (approx. 33%), followed by the anterior-most region of the prefrontal cortex or Brodmann area 10 (approx. 25%), and the lateral portions of the OFC or Brodmann area 47 (approx. 12%). From our analysis, it is difficult to conclude whether the detection of prediction errors in our study was specific to a certain OFC area, or whether different subregions were involved more than others during different types of stimuli/contexts processing.

To provide a more balanced interpretation of our findings, we incorporated a section in the “Discussion”, titled “Limitations and future directions” [page 32-33], which delves into the limitations of our study and lesion studies generally with respect to anatomical specificity and the challenge to establish large patient cohorts.

Reviewer #3 (Recommendations for the authors):The labeling of the Local+Global deviant ("xy|xY") seems to be wrong in paragraph 2 of the section "The Local-Global paradigm and procedures" and in figure 2.

We thank the reviewer for pointing this out. The labeling of the Local+Global deviant ("xy|xY") has been corrected in both paragraph 1 of the section "Materials and methods – The Local-Global paradigm and procedures" [page 37, paragraph 1] and in Figure 2 [page 10].

References

Alho, K., Woods, D. L., Algazi, A., Knight, R., and Näätänen, R. (1994). Lesions of frontal cortex diminish the auditory mismatch negativity. *Electroencephalography and clinical neurophysiology*, *91*(5), 353-362.

Bates, E., Wilson, S. M., Saygin, A. P., Dick, F., Sereno, M. I., Knight, R. T., and Dronkers, N. F. (2003). Voxel-based lesion–symptom mapping. *Nature neuroscience*, *6*(5), 448-450.

Bekinschtein, T. A., Dehaene, S., Rohaut, B., Tadel, F., Cohen, L., and Naccache, L. (2009). Neural signature of the conscious processing of auditory regularities. *Proceedings of the National Academy of Sciences*, *106*(5), 1672-1677.

Chao, L., Nielsen-Bohlman, L., and Knight, R. (1995). Auditory event-related potentials dissociate early and late memory processes. *Electroencephalography and Clinical Neurophysiology/Evoked Potentials Section*, *96*(2), 157-168.

Chao, Z. C., Takaura, K., Wang, L., Fujii, N., and Dehaene, S. (2018). Large-scale cortical networks for hierarchical prediction and prediction error in the primate brain. *Neuron*, *100*(5), 1252-1266. e1253.

Chennu, S., Noreika, V., Gueorguiev, D., Blenkmann, A., Kochen, S., Ibánez, A., Owen, A. M., and Bekinschtein, T. A. (2013). Expectation and attention in hierarchical auditory prediction. *Journal of Neuroscience*, *33*(27), 11194-11205.

Corbetta, M., Patel, G., and Shulman, G. L. (2008). The reorienting system of the human brain: from environment to theory of mind. *Neuron*, *58*(3), 306-324.

Corbetta, M., and Shulman, G. L. (2002). Control of goal-directed and stimulus-driven attention in the brain. *Nature reviews neuroscience*, *3*(3), 201-215.

Del Cul, A., Baillet, S., and Dehaene, S. (2007). Brain dynamics underlying the nonlinear threshold for access to consciousness. *PLoS biology*, *5*(10), e260.

Deouell, L. Y. (2007). The frontal generator of the mismatch negativity revisited. *Journal of Psychophysiology*, *21*(3-4), 188-203.

Donchin, E., and Coles, M. G. (1988). Is the P300 component a manifestation of context updating? *Behavioral and brain sciences*, *11*(3), 357-374.

Doricchi, F., Pinto, M., Pellegrino, M., Marson, F., Aiello, M., Campana, S., Tomaiuolo, F., and Lasaponara, S. (2021). Deficits of hierarchical predictive coding in left spatial neglect. *Brain communications*, *3*(2), fcab111.

Dürschmid, S., Edwards, E., Reichert, C., Dewar, C., Hinrichs, H., Heinze, H.-J., Kirsch, H. E., Dalal, S. S., Deouell, L. Y., and Knight, R. T. (2016). Hierarchy of prediction errors for auditory events in human temporal and frontal cortex. *Proceedings of the National Academy of Sciences*, *113*(24), 6755-6760.

El Karoui, I., King, J.-R., Sitt, J., Meyniel, F., Van Gaal, S., Hasboun, D., Adam, C., Navarro, V., Baulac, M., and Dehaene, S. (2015). Event-related potential, time-frequency, and functional connectivity facets of local and global auditory novelty processing: an intracranial study in humans. *Cerebral cortex*, *25*(11), 4203-4212.

Escera, C., and Corral, M. (2007). Role of mismatch negativity and novelty-P3 in involuntary auditory attention. *Journal of psychophysiology*, *21*(3-4), 251-264.

Fischer, C., Luauté, J., Adeleine, P., and Morlet, D. (2004). Predictive value of sensory and cognitive evoked potentials for awakening from coma. *Neurology*, *63*(4), 669-673.

Friston, K. (2005). A theory of cortical responses. *Philosophical transactions of the Royal Society B: Biological sciences*, *360*(1456), 815-836.

Garrido, M. I., Friston, K. J., Kiebel, S. J., Stephan, K. E., Baldeweg, T., and Kilner, J. M. (2008). The functional anatomy of the MMN: a DCM study of the roving paradigm. *Neuroimage*, *42*(2), 936-944.

Garrido, M. I., Kilner, J. M., Kiebel, S. J., and Friston, K. J. (2007). Evoked brain responses are generated by feedback loops. *Proceedings of the National Academy of Sciences*, *104*(52), 20961-20966.

Garrido, M. I., Kilner, J. M., Kiebel, S. J., and Friston, K. J. (2009). Dynamic causal modeling of the response to frequency deviants. *Journal of Neurophysiology*, *101*(5), 2620-2631.

Holroyd, C. B., and Coles, M. G. (2002). The neural basis of human error processing: reinforcement learning, dopamine, and the error-related negativity. *Psychological review*, *109*(4), 679.

Hämäläinen, M., Hari, R., Ilmoniemi, R. J., Knuutila, J., and Lounasmaa, O. V. (1993). Magnetoencephalography—theory, instrumentation, and applications to noninvasive studies of the working human brain. *Reviews of modern Physics*, *65*(2), 413.

Kakade, S., and Dayan, P. (2002). Dopamine: generalization and bonuses. *Neural Networks*, *15*(4-6), 549-559.

Knight, R. T. (1984). Decreased response to novel stimuli after prefrontal lesions in man. *Electroencephalography and Clinical Neurophysiology/Evoked Potentials Section*, *59*(1), 9-20.

Knight, R. T. (1997). Distributed cortical network for visual attention. *Journal of Cognitive Neuroscience*, *9*(1), 75-91.

Knight, R. T., and Scabini, D. (1998). Anatomic bases of event-related potentials and their relationship to novelty detection in humans. *Journal of clinical neurophysiology*, *15*(1), 3-13.

Kompus, K., Volehaugen, V., Todd, J., and Westerhausen, R. (2020). Hierarchical modulation of auditory prediction error signaling is independent of attention. *Cognitive neuroscience*, *11*(3), 132-142.

Kutas, M., Kiang, M., and Sweeney, K. (2012). Potentials and Paradigms: Event‐Related Brain Potentials and Neuropsychology. *The handbook of the neuropsychology of language*, *1*, 543-564.

Liaukovich, K., Ukraintseva, Y., and Martynova, O. (2022). Implicit auditory perception of local and global irregularities in passive listening condition. *Neuropsychologia*, *165*, 108129.

Lieder, F., Daunizeau, J., Garrido, M. I., Friston, K. J., and Stephan, K. E. (2013). Modelling trial-by-trial changes in the mismatch negativity. *PLoS computational biology*, *9*(2), e1002911.

Lorca-Puls, D. L., Gajardo-Vidal, A., White, J., Seghier, M. L., Leff, A. P., Green, D. W., Crinion, J. T., Ludersdorfer, P., Hope, T. M., and Bowman, H. (2018). The impact of sample size on the reproducibility of voxel-based lesion-deficit mappings. *Neuropsychologia*, *115*, 101-111.

Løvstad, A., and Cawley, P. (2011). The reflection of the fundamental torsional guided wave from multiple circular holes in pipes. *Ndt and E International*, *44*(7), 553-562.

Løvstad, M., Funderud, I., Lindgren, M., Endestad, T., Due-Tønnessen, P., Meling, T., Voytek, B., Knight, R. T., and Solbakk, A.-K. (2012). Contribution of subregions of human frontal cortex to novelty processing. *Journal of Cognitive Neuroscience*, *24*(2), 378-395.

Naccache, L., Puybasset, L., Gaillard, R., Serve, E., and Willer, J.-C. (2004). Auditory mismatch negativity is a good predictor of awakening in comatose patients: a fast and reliable procedure. *Clinical neurophysiology: official journal of the International Federation of Clinical Neurophysiology*, *116*(4), 988-989.

Nieuwenhuis, S., Aston-Jones, G., and Cohen, J. D. (2005). Decision making, the P3, and the locus coeruleus--norepinephrine system. *Psychological bulletin*, *131*(4), 510.

Noonan, M., Kolling, N., Walton, M., and Rushworth, M. (2012). Re‐evaluating the role of the orbitofrontal cortex in reward and reinforcement. *European Journal of Neuroscience*, *35*(7), 997-1010.

Nourski, K. V., Steinschneider, M., Rhone, A. E., Kawasaki, H., Howard III, M. A., and Banks, M. I. (2018). Processing of auditory novelty across the cortical hierarchy: An intracranial electrophysiology study. *Neuroimage*, *183*, 412-424.

Näätänen, R., Pakarinen, S., Rinne, T., and Takegata, R. (2004). The mismatch negativity (MMN): towards the optimal paradigm. *Clinical neurophysiology*, *115*(1), 140-144.

Näätänen, R., Tervaniemi, M., Sussman, E., Paavilainen, P., and Winkler, I. (2001). ‘Primitive intelligence’in the auditory cortex. *Trends in neurosciences*, *24*(5), 283-288.

Padoa-Schioppa, C., and Assad, J. A. (2006). Neurons in the orbitofrontal cortex encode economic value. *Nature*, *441*(7090), 223-226.

Pegado, F., Bekinschtein, T., Chausson, N., Dehaene, S., Cohen, L., and Naccache, L. (2010). Probing the lifetimes of auditory novelty detection processes. *Neuropsychologia*, *48*(10), 3145-3154.

Phillips, H. N., Blenkmann, A., Hughes, L. E., Bekinschtein, T. A., and Rowe, J. B. (2015). Hierarchical organization of frontotemporal networks for the prediction of stimuli across multiple dimensions. *Journal of Neuroscience*, *35*(25), 9255-9264.

Phillips, H. N., Blenkmann, A., Hughes, L. E., Kochen, S., Bekinschtein, T. A., and Rowe, J. B. (2016). Convergent evidence for hierarchical prediction networks from human electrocorticography and magnetoencephalography. *cortex*, *82*, 192-205.

Polich, J. (2007). Updating P300: an integrative theory of P3a and P3b. *Clinical neurophysiology*, *118*(10), 2128-2148.

Rosburg, T., Trautner, P., Dietl, T., Korzyukov, O. A., Boutros, N. N., Schaller, C., Elger, C. E., and Kurthen, M. (2005). Subdural recordings of the mismatch negativity (MMN) in patients with focal epilepsy. *Brain*, *128*(4), 819-828.

Rugg, M. D. (1995). Event-related potential studies of human memory.

Schomaker, J., Roos, R., and Meeter, M. (2014). Expecting the unexpected: The effects of deviance on novelty processing. *Behavioral neuroscience*, *128*(2), 146.

Schultz, W. (2015). Neuronal reward and decision signals: from theories to data. *Physiological reviews*, *95*(3), 853-951.

Schultz, W., and Dickinson, A. (2000). Neuronal coding of prediction errors. *Annual review of neuroscience*, *23*(1), 473-500.

Sculthorpe, L. D., Stelmack, R. M., and Campbell, K. B. (2009). Mental ability and the effect of pattern violation discrimination on P300 and mismatch negativity. *Intelligence*, *37*(4), 405-411.

Sergent, C., Baillet, S., and Dehaene, S. (2005). Timing of the brain events underlying access to consciousness during the attentional blink. *Nature neuroscience*, *8*(10), 1391-1400.

Seymour, B., O'Doherty, J. P., Dayan, P., Koltzenburg, M., Jones, A. K., Dolan, R. J., Friston, K. J., and Frackowiak, R. S. (2004). Temporal difference models describe higher-order learning in humans. *Nature*, *429*(6992), 664-667.

Stalnaker, T. A., Cooch, N. K., and Schoenbaum, G. (2015). What the orbitofrontal cortex does not do. *Nature neuroscience*, *18*(5), 620-627.

Strauss, M., Sitt, J. D., King, J.-R., Elbaz, M., Azizi, L., Buiatti, M., Naccache, L., Van Wassenhove, V., and Dehaene, S. (2015). Disruption of hierarchical predictive coding during sleep. *Proceedings of the National Academy of Sciences*, *112*(11), E1353-E1362.

Sul, J. H., Kim, H., Huh, N., Lee, D., and Jung, M. W. (2010). Distinct roles of rodent orbitofrontal and medial prefrontal cortex in decision making. *Neuron*, *66*(3), 449-460.

Swick, D. (2005). 13 ERPs in Neuropsychological Populations. *Event-related potentials: A methods handbook*, 299.

Swaab, T. Y. (1998). Event-related potentials in cognitive neuropsychology: Methodological considerations and an example from studies of aphasia. *Behavior Research Methods, Instruments, and Computers*, *30*(1), 157-170.

Tiitinen, H., May, P., Reinikainen, K., and Näätänen, R. (1994). Attentive novelty detection in humans is governed by pre-attentive sensory memory. *Nature*, *372*(6501), 90-92.

Tobler, P. N., O’Doherty, J. P., Dolan, R. J., and Schultz, W. (2006). Human neural learning depends on reward prediction errors in the blocking paradigm. *Journal of Neurophysiology*, *95*(1), 301-310.

Tremblay, L., and Schultz, W. (1999). Relative reward preference in primate orbitofrontal cortex. *Nature*, *398*(6729), 704-708.

Uhrig, L., Dehaene, S., and Jarraya, B. (2014). A hierarchy of responses to auditory regularities in the macaque brain. *Journal of Neuroscience*, *34*(4), 1127-1132.

Ungan, P., Karsilar, H., and Yagcioglu, S. (2019). Pre-attentive mismatch response and involuntary attention switching to a deviance in an earlier-than-usual auditory stimulus: an ERP study. *Frontiers in Human Neuroscience*, *13*, 58.

Wacongne, C., Labyt, E., van Wassenhove, V., Bekinschtein, T., Naccache, L., and Dehaene, S. (2011). Evidence for a hierarchy of predictions and prediction errors in human cortex. *Proceedings of the National Academy of Sciences*, *108*(51), 20754-20759.

Walton, M. E., Behrens, T. E., Buckley, M. J., Rudebeck, P. H., and Rushworth, M. F. (2010). Separable learning systems in the macaque brain and the role of orbitofrontal cortex in contingent learning. *Neuron*, *65*(6), 927-939.

Walton, M. E., Behrens, T. E., Noonan, M. P., and Rushworth, M. F. (2011). Giving credit where credit is due: orbitofrontal cortex and valuation in an uncertain world. *Annals of the New York Academy of Sciences*, *1239*(1), 14-24.

Wessel, J. R., Danielmeier, C., Morton, J. B., and Ullsperger, M. (2012). Surprise and error: common neuronal architecture for the processing of errors and novelty. *Journal of Neuroscience*, *32*(22), 7528-7537.

Wessel, J. R., Klein, T. A., Ott, D. V., and Ullsperger, M. (2014). Lesions to the prefrontal performance-monitoring network disrupt neural processing and adaptive behaviors after both errors and novelty. *Cortex*, *50*, 45-54.

Yamaguchi, S., and Knight, R. (1991). Anterior and posterior association cortex contributions to the somatosensory P300. *Journal of Neuroscience*, *11*(7), 2039-2054.